# BNP facilitates NMB-encoded histaminergic itch via NPRC-NMBR crosstalk

Qing-Tao Meng[1†‡], Xian-Yu Liu[1,2†], Xue-Ting Liu[1,3,4†], Juan Liu[1,2†], Admire Munanairi[1,2], Devin M Barry[1,2], Benlong Liu[1,2], Hua Jin[1,2§], Yu Sun[1#], Qianyi Yang[1,2], Fang Gao[1,2], Li Wan[1,5], Jiahang Peng[1,2], Jin-Hua Jin[1¶], Kai-Feng Shen[1**], Ray Kim[1††], Jun Yin[1,2], Ailin Tao[4], Zhou-Feng Chen[1,2,3,4,6,7*]

[1]Center for the Study of Itch and Sensory Disorders, Washington University School of Medicine, St Louis, United States; [2]Departments of Anesthesiology, Washington University School of Medicine, St Louis, United States; [3]Developmental Biology, Washington University School of Medicine, St. Louis, United States; [4]The Second Affiliated Hospital, The State Key Laboratory of Respiratory Disease, Guangdong Provincial Key Laboratory of Allergy & Clinical Immunology, Guangzhou Medical University, Guangzhou, China; [5]Department of Pain, Guangzhou Medical University, Guangzhou, China; [6]Departments of Medicine, Washington University School of Medicine, St. Louis, United States; [7]Departments of Psychiatry, Washington University School of Medicine, St. Louis, United States

*For correspondence: chenz@wustl.edu

†These authors contributed equally to this work

Present address: ‡Department of Anesthesiology, Renmin Hospital of Wuhan University, Wuhan, China; §The First Hospital of Yunnan Province, Kunming, Yunnan, China; #Department of Anesthesiology, Shanghai Ninth People's Hospital, Shanghai Jiaotong University School of Medicine, Shanghai, China; ¶Department of Anesthesiology, Chinese Academy of Medical Sciences Plastic Surgery Hospital, Beijing, Beijing, China; **Department of Neurosurgery, Xinqiao Hospital, Third Military Medical University, Chongqing, China; ††Department of Genetics, Yale University School of Medicine, New Haven, China

Competing interest: The authors declare that no competing interests exist.

## Abstract

Histamine-dependent and -independent itch is conveyed by parallel peripheral neural pathways that express gastrin-releasing peptide (GRP) and neuromedin B (NMB), respectively, to the spinal cord of mice. B-type natriuretic peptide (BNP) has been proposed to transmit both types of itch via its receptor NPRA encoded by *Npr1*. However, BNP also binds to its cognate receptor, NPRC encoded by *Npr3* with equal potency. Moreover, natriuretic peptides (NP) signal through the $G_i$-couped inhibitory cGMP pathway that is supposed to inhibit neuronal activity, raising the question of how BNP may transmit itch information. Here, we report that *Npr3* expression in laminae I-II of the dorsal horn partially overlaps with NMB receptor (NMBR) that transmits histaminergic itch via $G_q$-couped PLCβ-$Ca^{2+}$ signaling pathway. Functional studies indicate that NPRC is required for itch evoked by histamine but not chloroquine (CQ), a nonhistaminergic pruritogen. Importantly, BNP significantly facilitates scratching behaviors mediated by NMB, but not GRP. Consistently, BNP evoked $Ca^{2+}$ responses in NMBR/NPRC HEK 293 cells and NMBR/NPRC dorsal horn neurons. These results reveal a previously unknown mechanism by which BNP facilitates NMB-encoded itch through a novel NPRC-NMBR cross-signaling in mice. Our studies uncover distinct modes of action for neuropeptides in transmission and modulation of itch in mice.

## Editor's evaluation

The study by Meng et al., reveals how two distinct neuropeptide signals intersect to drive histaminergic itch. They find that the neuropeptides B-type natriuretic peptide (BNP) and neuromedin B (NMB) crosstalk in the spinal cord, whereby BNP facilitates itch behaviors driven by NMB via coupling between the receptors NPRC and NMBR. These results demonstrate a mechanism underlying spinal cord itch processing.

## Introduction

How itch and pain information is encoded and transmitted has been subjected to numerous studies for more than a century (*Chen, 2021*). There is increasing evidence indicating the pivotal roles of neuropeptides in the coding of itch information in primary sensory neurons (*Chen, 2021*). A pruritogenic stimulus activates skin, immune, and nerve cells, or an inflammatory response, which provokes the release of itch-specific neuropeptides from primary afferents to activate G-protein-coupled receptors (GPCRs) in the spinal cord (*Chen, 2021*; *Wang and Kim, 2020*). Notably, gastrin-releasing peptide (GRP) and neuromedin B (NMB), two mammalian neuropeptides, have been shown to encode nonhistaminergic itch and histaminergic itch, respectively (*Akiyama et al., 2014*; *Barry et al., 2020*; *Sun and Chen, 2007*; *Wan et al., 2017*; *Zhao et al., 2014b*). Moreover, murine GRP-GRPR signaling is important for the development of contact dermatitis-induced itch (*Chen et al., 2020*; *Liu et al., 2020*; *Shiratori-Hayashi et al., 2015*; *Zhao et al., 2013*). These findings are in accordance with human and animal studies showing that histaminergic and nonhistaminergic itch is transmitted through parallel primary afferent pathways (*Chen, 2021*; *Johanek et al., 2007*; *Namer et al., 2008*; *Roberson et al., 2013*; *Wilson et al., 2011*).

B-type or brain natriuretic peptide (BNP), encoded by the gene *Nppb,* has been implicated in itch at discrete regions, including skin cells, sensory neurons, and spinal cord (*Liu et al., 2020*; *Meng et al., 2018*; *Mishra and Hoon, 2013*; *Solinski et al., 2019*). The natriuretic peptide (NP) family also consists of atrial (ANP) and C-type natriuretic peptides (CNP) (*Potter et al., 2006*). BNP binds to both NPRA and NPRC, encoded by *Npr1* and *Npr3*, respectively, with equality affinity, but not NPRB, while ANP also binds NPRA directly, resulting in the elevation of the second message cyclic GMP concentration (*Figure 1A*; *Potter et al., 2006*). Although NPRC is considered to function as a clearance or silent receptor (*Maack et al., 1987*), it can also mediate guanylyl cyclase (GC) receptor-coupled $G_{\alpha i}$ signaling under certain physiological conditions (*Anand-Srivastava, 2005*). BNP-NPRA signaling was initially proposed as an itch-specific pathway responsible for transmitting both histamine- and CQ-evoked itch that acts upstream of GRP-GRPR signaling (*Huang et al., 2018*; *Mishra and Hoon, 2013*). However, that BNP transmits all types of itch is at odds with the fact that GRP is required only for nonhistaminergic itch. Further, genetic ablation of spinal *Grp* neurons fails to impact itch behaviors (*Barry et al., 2020*), indicating that spinal *Grp* neurons do not constitute a functional circuit for itch. Recent studies have shown that BNP-NPRA signaling is involved in histaminergic itch as well as chronic itch in mice which comprises the histaminergic component (*Liu et al., 2020*; *Solinski et al., 2019*). Given that BNP can bind both NPRA and NPRC, two cognate receptors for BNP (*Figure 1A*), the relationship between NPRA/NPRC and NMBR/GRPR and the role of NPRC in itch transmission remains undefined. Considering that the GC-cGMP signal transduction pathway mediated by BNP is inhibitory (*Potter et al., 2006*) and that BNP-NPRA/NPRC signaling may exert an inhibitory rather than excitatory function, analogous to $G_{\alpha i}$ protein-coupled signaling, it is paradoxical that BNP would transmit rather than inhibiting itch information.

In the present study, we have examined these open questions using a combination of RNA-scope ISH, genetic knockout (KO) mice, spinal siRNA knockdown, cell ablation, calcium imaging, pharmacological and optogenetic approaches. We found that NPRC, rather than NPRA, is a major functional receptor for BNP in the spinal cord, and BNP facilitates histamine-evoked itch through NPRC-NMBR crosstalk. Importantly, our studies confirmed that BNP is an inhibitory neuropeptide that alone fails to evoke $Ca^{2+}$ response and itch-related scratching behavior, in contrast to GRP and NMB; However, BNP becomes excitatory by facilitating NMB-mediated itch transmission. Thus. distinct modes of action for neuropeptides coordinate itch transmission in the spinal cord.

## Results

### Expression of NP Receptors in the spinal cord

As ANP also binds NPRA at high affinity, we tested whether intrathecal injection (i.t.) of ANP could induce scratching behavior and found that ANP failed to induce robust scratching behaviors at the dose of 1–20 µg (equivalent to 6–120 µM, *Figure 1—figure supplement 1A*). Among three NPs, interestingly, only BNP facilitates histamine itch (*Figure 1—figure supplement 1B*). BNP evoked dose-related scratching behavior (1–5 µg that is equivalent to 30–150 µM) with a peak scratching number of 74 ± 16.2 (*Figure 1B*), consistent with previous reports (*Kiguchi et al., 2016*; *Liu et al.,*

**eLife digest** An itch is a common sensation that makes us want to scratch. Most short-term itches are caused by histamine, a chemical that is released by immune cells following an infection or in response to an allergic reaction. Chronic itching, on the other hand, is not usually triggered by histamine, and is typically the result of neurological or skin disorders, such as atopic dermatitis.

The sensation of itching is generated by signals that travel from the skin to nerve cells in the spinal cord. Studies in mice have shown that the neuropeptides responsible for delivering these signals differ depending on whether or not the itch involves histamine: GRPs (short for gastrin-releasing proteins) convey histamine-independent itches, while NMBs (short for neuromedin B) convey histamine-dependent itches.

It has been proposed that another neuropeptide called BNP (short for B-type natriuretic peptide) is able to transmit both types of itch signals to the spinal cord. But it remains unclear how this signaling molecule is able to do this.

To investigate, Meng, Liu, Liu, Liu et al. carried out a combination of behavioral, molecular and pharmacological experiments in mice and nerve cells cultured in a laboratory. The experiments showed that BNP alone cannot transmit the sensation of itching, but it can boost itching signals that are triggered by histamine.

It is widely believed that BNP activates a receptor protein called NPRA. However, Meng et al. found that the BNP actually binds to another protein which alters the function of the receptor activated by NMBs. These findings suggest that BNP modulates rather than initiates histamine-dependent itching by enhancing the interaction between NMBs and their receptor.

Understanding how itch signals travel from the skin to neurons in the spinal cord is crucial for designing new treatments for chronic itching. The work by Meng et al. suggests that treatments targeting NPRA, which was thought to be a key itch receptor, may not be effective against chronic itching, and that other drug targets need to be explored.

*2014*). However, these doses are much higher than endogenous concentrations of ligands that should be within the nanomolar range, implying that scratching behaviors evoked by BNP reflect a non-specific pharmacological artifact. Time-course analysis showed that scratching behavior was delayed by 20–30 min after BNP injections as described without isoflurane treatment (*Kiguchi et al., 2016*; *Liu et al., 2014*; *Figure 1C*), which is distinct from the rapid onset of scratching response evoked by i.t. GRP or NMB (*Figure 1—figure supplement 2H*). The use of isoflurane for anesthesia could result in complex effects on neural circuits, especially inhibitory circuits (*Constantinides and Murphy, 2016*). We compared the effect of i.t. BNP on awake and isoflurane-anesthetized animals and found that isoflurane pretreatment significantly enhanced BNP-induced scratching behaviors in the first 10 min (*Figure 1—figure supplement 2G*). This suggests that induced scratching behavior is an indirect rather than direct effect of BNP with isoflurane treatment, consistent with the fact that NPRA/NPRC are inhibitory receptors. Recent studies showed that *Npr1* is widespread in the dorsal horn (*Fatima et al., 2019*), in marked contrast to lamina II specific *Grp* expression (*Barry et al., 2020*). Consistently, single nucleus RNA sequencing (snRNA-seq) from isolated spinal cord neurons found only a partial overlap (20 ~ 30%) of *Npr1* and *Grp* (*Sathyamurthy et al., 2018*). To visualize the distribution of NP receptors in the spinal cord, we performed RNAscope in situ hybridization (ISH) and found that all three NP receptors are expressed in the dorsal horn of the spinal cord (*Figure 1D-N* and Figure 5A-C). Consistent with a previous ISH study (*Barry et al., 2020*; *Fatima et al., 2019*; *Mishra and Hoon, 2013*), *Npr1*$^+$ neurons are distributed in a gradient manner with higher intensity throughout laminae I-IV and are rather sparse in the deep dorsal horn (*Figure 1D* and 5A). Remarkably, *Npr3*$^+$ + are predominantly restricted to lamina I-II (*Figure 1H, J, L, N*, 5C) with 64.6% being excitatory neurons and 32.0% being inhibitory neurons (*Figure 1J–M*). In contrast, *Npr2* expression is homogenous throughout the spinal cord dorsal horn, implying that *Npr2* lacks modality-specific function (Figure 5B). RNAscope ISH showed minimal overlapping expression between *Npr1* and *Grpr* (*Figure 1D and E*). However, approximately 30% (31/103) of *Npr1*$^+$ + in laminae I-II of the dorsal horn express *Grp* (*Figure 1—figure supplement 1E, F*), and this number is further reduced to 23% (31/132) (data not shown) when all *Npr1*$^+$ neurons in the dorsal horn are counted. Moreover, *Npr1* and *Nmbr* minimally

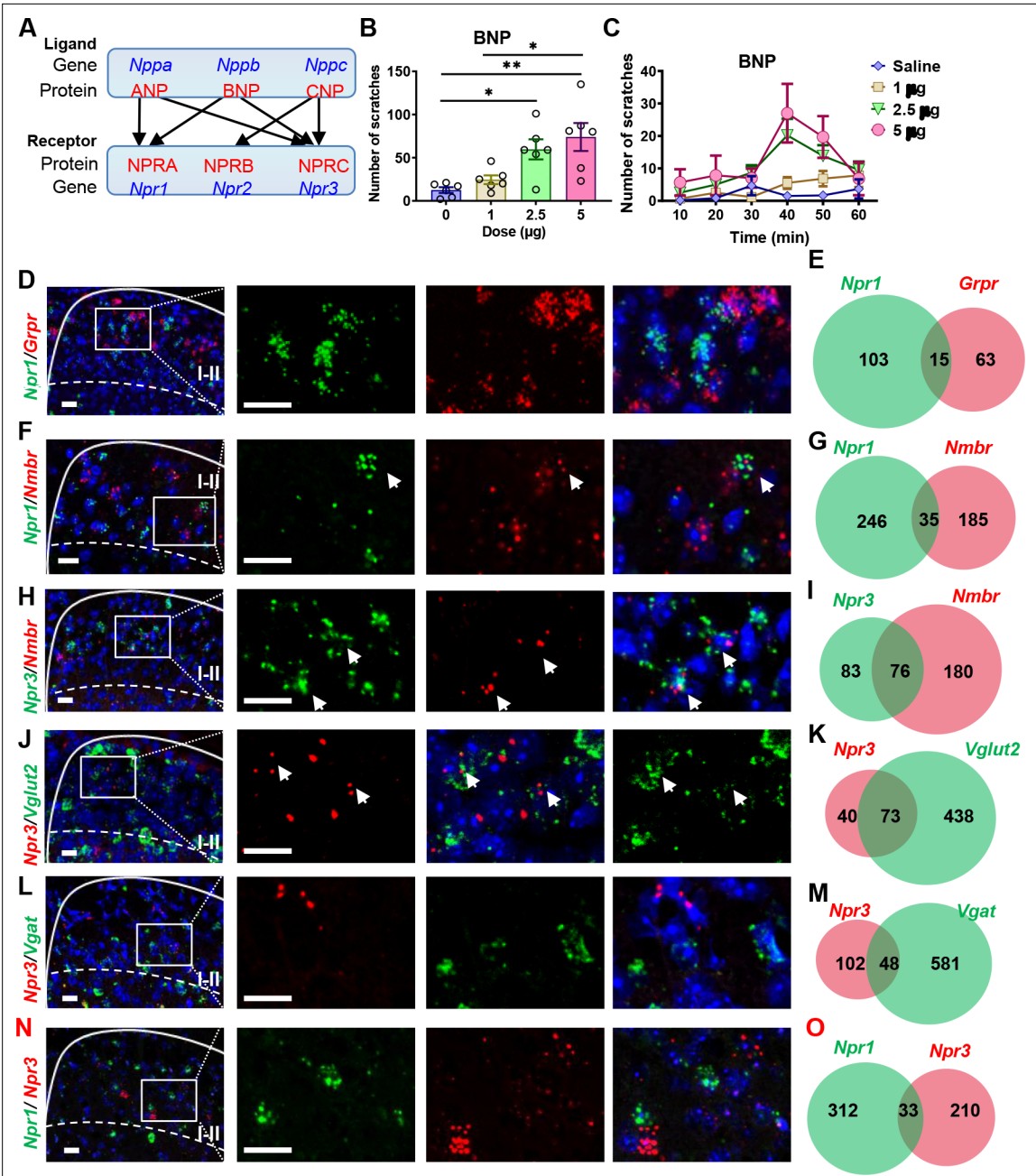

**Figure 1.** Expression of *Npr1*, 2,and *3* and other molecular markers in the spinal cord. (**A**) Diagram shows crosstalk between NPs and NP receptors. BNP can bind NPRA and NPRC. (**B**) BNP dose-dependently evoked scratching behaviors 60 min after i.t. injection. n = 6. *p < 0.05, **p < 0.01, one-way ANOVA followed by Tukey's test. (**C**) Time-course of scratching behaviors induced by different doses of BNP shows a delayed onset of scratching responses. (**D, F, H, J, L, N**) Images of double RNAscope ISH showing that the overlapping expression of *Npr1* (green) with *Grpr* (red) (**D**), *Nmbr* (**F**), of *Npr3* (green) with *Nmbr* (red) (**H**), *Npr3* (red) with *Vglut2* (green) (**J**), *Vgat* (green) (**L**), or *Npr1* (green) (**N**) in laminae I-II of the dorsal horn. Dashed white lines divide laminae I-II from III. White boxes are shown at higher magnification in the right panel. Arrows indicate double-positive neurons. E, G, I, K, M, O, Venn diagrams showing the overlap between *Npr1* and *Grpr* (**E**), *Nmbr* (**G**), between *Npr3* and *Nmbr* (**I**), *Vglut2* (**K**), *Vgat* (**M**) or *Npr1* (**O**). n = 10–15 sections from 3 mice. Scale bar, 20 μm in **D – N**.

The online version of this article includes the following source data and figure supplement(s) for figure 1:

**Source data 1.** BNP dose-dependently evoked scratching behaviors and showed a delayed onset of scratching responses.

**Figure supplement 1.** Failure of ANP and CNP in facilitating histamine itch.

**Figure supplement 1—source data 1.** Failure of ANP and CNP in facilitating histamine itch.

*Figure 1 continued on next page*

Figure 1 continued

**Figure supplement 2.** Normal innervation of primary afferents in *Npr1* KO mice and WT mice.

**Figure supplement 2—source data 1.** Time course of NMB and GRP evoked scratching behavior.

overlap in the dorsal horn (*Figure 1F and G*), excluding the likelihood of NPRA-NMBR crosstalk. *Npr1* and *Npr3* also showed minimal overlapping expression in the spinal cord (*Figure 1N and O*). By contrast, approximately 47.8% of $Npr3^+$ + express *Nmbr* (*Figure 1H,I*), raising the possibility that NPRC is involved in crosstalk with NMBR.

## NPRA and NPRC are important for acute itch

To examine the role of NP receptors in acute itch behavior, we first analyzed the phenotype of *Npr1* knockout (KO) mice (*Oliver et al., 1997*). A previous study showing that CNP-NPRB is essential for axonal bifurcation of DRG neurons in the developing spinal cord (*Schmidt et al., 2009*) prompted us to evaluate the innervation of primary afferents in the spinal cord of *Npr1* KO mice. We found that innervations of peptidergic $CGRP^+$ and non-peptidergic $IB4^+$ primary afferents in the superficial dorsal horn of *Npr1* KO mice are comparable with wild-type (WT) littermates (*Figure 1—figure supplement 2A*). The innervations of $TRPV1^+$, $GRP^+$, and $SP^+$ primary afferents are also comparable between WT and *Npr1* KO mice (*Figure 1—figure supplement 2B-D*), indicating that NPRA is dispensable for the innervation of primary afferents.

GRP and NMB have been implicated in nonhistaminergic and histaminergic itch, respectively (*Sun et al., 2009*; *Wan et al., 2017*; *Zhao et al., 2014b*). To examine whether GRPR and NMBR function normally in the absence of NPRA, we compared the scratching behaviors between *Npr1* KO mice and their WT littermates after i.t. injection of GRP or NMB and found no significant differences in their responses to either GRP or NMB between the groups (*Figure 2A*). However, *Npr1* KO mice showed significantly impaired scratching responses to intradermal (i.d.) injection of histamine and chloroquine (CQ), archetypal pruritogens for histaminergic and nonhistaminergic itch, respectively (*Sun and Chen, 2007*), as compared with WT littermates (*Figure 2B*).

The highly restricted expression of NPRC in lamina II and its high binding affinity to BNP prompted us to examine the role of NPRC in itch. However, *Npr3* KO mice showed severe skeletal abnormalities, resulting in the failure of most KO mice to survive to adult stage for behavioral analysis (*Matsukawa et al., 1999*). To determine whether the impaired scratching response of the global *Npr1* KO mice could have resulted from the *Npr1* deficiency in the spinal cord, DRGs where *Npr1* is also expressed (*Zhang et al., 2010*), or skin cells (*Meng et al., 2018*), we knocked down *Npr1-3* either individually or in combination in C57Bl/6 mice using sequence-specific siRNA. I.t. *Npr1* siRNA treatment significantly attenuated the scratching behavior evoked by histamine and CQ (*Figure 2C and D*), whereas *Npr3* siRNA treatment selectively attenuated histamine, but not CQ itch (*Figure 2C and D*). *Npr2* siRNA had no effect on CQ and histamine itch, making it unlikely to be involved in itch transmission (*Figure 2C and D*). The effect of the knock-down of target mRNA in the spinal cord and DRGs was verified using real-time RT-PCR (*Figure 2E and F*). These results revealed that *Npr1* and *Npr3* are differentially required for acute itch behavior at the spinal level. Further, we infer that there is little functional compensation among the three NP receptors.

## BNP facilitates NMB-mediated histamine itch

The slow onset of scratching behavior elicited by BNP, even at a high dose (150 µM), in the first 30 min contrasts sharply with rapid onset of GRP/NMB-induced scratching behavior (*Figure 1—figure supplement 2H*), implying that direct activation of NPRA/NPRC itself is insufficient to initiate scratching response. This raises the question as to the specific role BNP may play in the first phase, which is physiologically relevant to the histamine and CQ itch that usually occurs within this period. We suspected that BNP may play a modulatory function in acute itch behavior in a manner resembling the role of serotonin in itch modulation (*Zhao et al., 2014a*). To test this, we pre-treated mice with BNP at a lower dose (30 µM, i.t.) followed by i.d. histamine at a dose of 100 µg that is insufficient to induce robust scratching behaviors. The low-dose effect was similar to the saline control, which enabled us to examine the facilitatory effect, rather than the additive effect (*Figure 3A–D*). At 30 µM, BNP failed to induce scratching behaviors (*Figure 1B*). Strikingly, histamine-induced scratching responses were

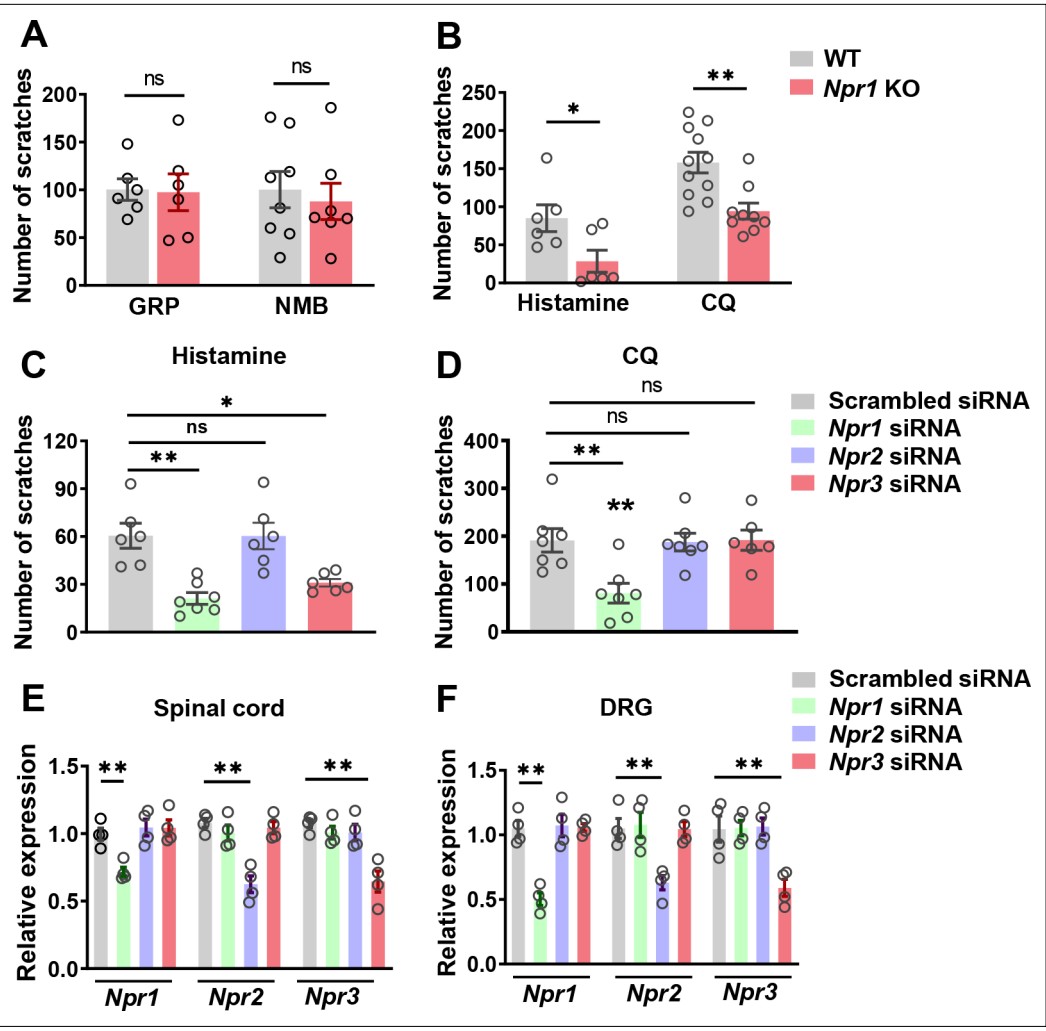

**Figure 2.** NPRA and NPRC are involved in acute itch. (**A**) *Npr1* KO mice and their WT littermates showed comparable scratching behaviors in response to GRP (0.05 nmol, i.t.) and NMB (0.5 nmol, i.t.). n = 6–8. (**B**) *Npr1* KO mice showed significantly reduced scratching behavior elicited by histamine (200 μg, i.d.) and CQ (200 μg, i.d.). n = 9–11. *p < 0.05, **p < 0.01, unpaired t test. (**C, D**) Mice treated with *Npr1* siRNA showed significantly reduced scratching responses to histamine (**C**), CQ (**D**), wherea mice treated with *Npr3* siRNA displayed deficits only in histamine (**C**) but not CQ itch (**D**). n = 6–7. *p < 0.05, **p < 0.01, one-way ANOVA followed by Dunnett's test. (**E, F**) Real-time PCR confirmed the reduced *Npr1-3* expression by *Npr1*, *Npr2*, and *Npr3* siRNA knockdown in the spinal cord (**E**) and DRG (**F**). n = 4. **p < 0.01, one-way ANOVA followed by Dunnett's test. Values are presented as mean ± SEM.

The online version of this article includes the following source data for figure 2:

**Source data 1.** NPRA and NPRC are involved in acute itch.

significantly enhanced with BNP pretreatment compared with saline control (*Figure 3A*). BNP also produced similar potentiating effects on CQ-induced scratching responses (*Figure 3B*). Since NMB is required for histamine itch via NMBR exclusively (*Wan et al., 2017*; *Zhao et al., 2014b*), we tested the possibility that BNP may facilitate histamine itch by modulating NMBR function. At 0.05 nmol, i.t. NMB itself could not induce significant scratching behavior (*Figure 3C*). However, co-injection of BNP (30 μM) and NMB (0.05 nmol) markedly increased NMB-induced scratching behavior compared with that of mice receiving only NMB (*Figure 3C*). Importantly, BNP failed to potentiate scratching behaviors induced by GRP (0.01 nmol, i.t.) (*Figure 3D*).

Next we assessed whether BNP may function upstream or independently of GRPR to modulate itch by comparing the facilitatory effect of BNP on histamine itch between *Grpr* KO and WT mice. If BNP acts upstream of or depends on GRPR, BNP may fail to potentiate histamine itch in *Grpr* KO mice. We

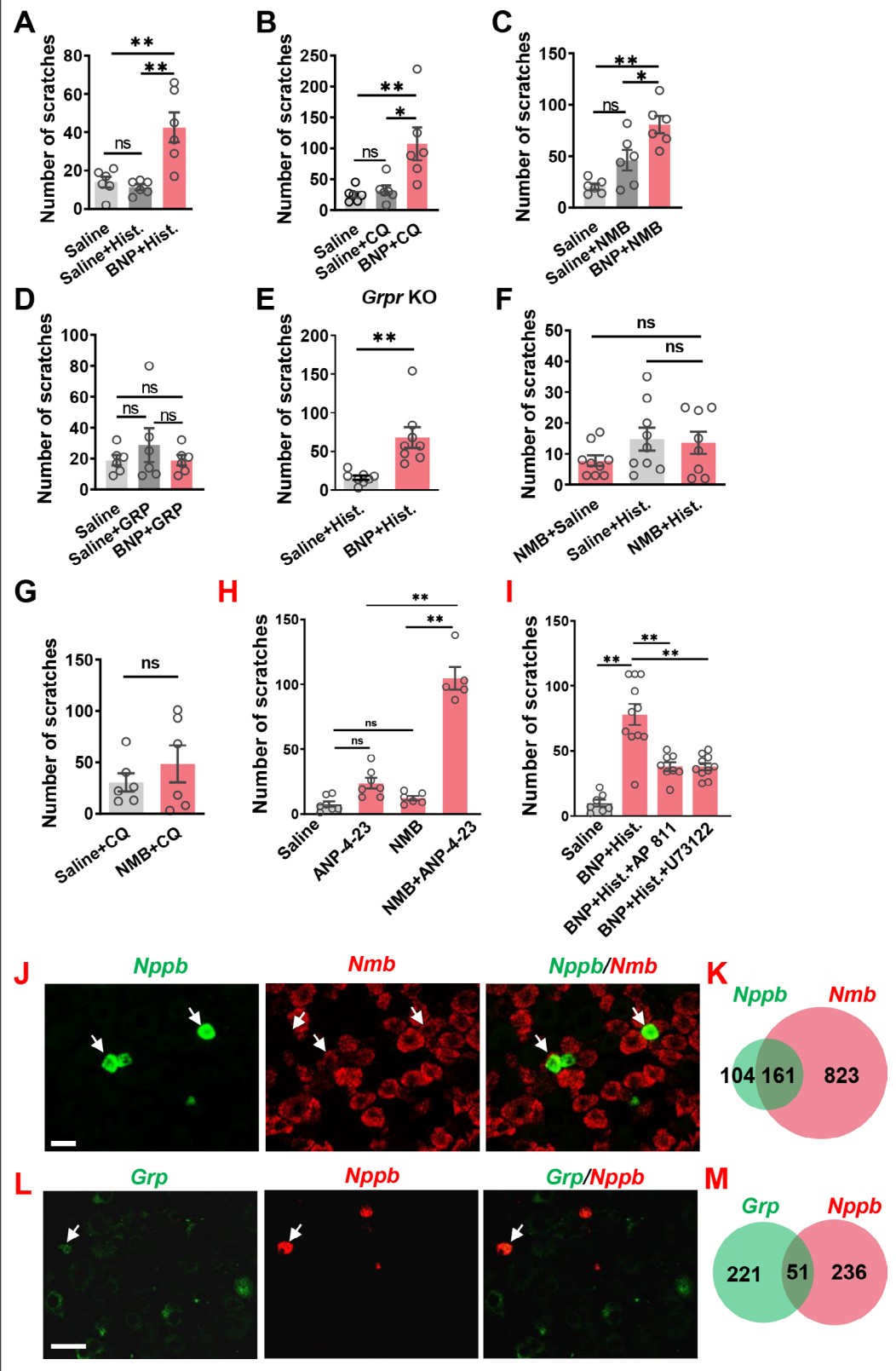

**Figure 3.** BNP facilitates histamine itch. (**A**) Pre-injection of BNP (30 μM, i.t.) for 1 min significantly enhanced scratching behavior evoked by i.d. injection of histamine (Hist.) (100 μg). n = 6. (**B**) Scratching behavior evoked by i.d. injection of CQ (50 μg, i.d.) was significantly enhanced by pre-injection of BNP for 1 min. n = 6. (**C, D**) Co-injection of 1 μg BNP (30 μM, i.t.) facilitated scratching behaviors evoked by NMB (0.05 nmol, i.t.) (**C**) but not GRP

*Figure 3 continued on next page*

*Figure 3 continued*

(0.01 nmol) (**D**). n = 6. (**E**) Pre-injection of 1 µg BNP (30 µM, i.t.) for 1 min significantly enhanced scratching behavior evoked by i.d. injection of histamine (100 µg) in *Grpr* KO mice. n = 8. (**F, G**) Pre-injection of NMB (0.05 nmol, i.t.) had no effect on scratching behavior induced by histamine (**F**) or CQ (**G**). Note that NMB barely evoked scratching bouts. n = 6. (**H**), NPRC agonist ANP-4–23 (6 nmol, i.t.) facilitates NMB (0.005 nmol, i.t.) induced scratching behavior. n = 5–9. (**I**), Histamine (25 µg, i.d.)-induced scratching behavior facilitated by BNP (30 µM, i.t.) was attenuated with AP 811 (10 µM, i.t.) or U 73122 (13.5 nmol, i.t.) treatment. n = 6–11. (**I–K**) Double RNAScope ISH images (**J and L**) and Venn diagrams (**K and M**) showing 60% of *Nppb* neurons co-express *Nmb* (**J and K**), but little *Grp* in DRGs (**L and M**). Values are presented as mean ± SEM, *p < 0.05, **p < 0.01, unpaired t test in (**A–E**), one-way ANOVA in (**F and G**). Scale bar, 20 µm in **J**, 50 µm in **L**.

The online version of this article includes the following source data for figure 3:

**Source data 1.** BNP facilitates histamine itch.

found that BNP similarly potentiated histamine itch in *Grpr* KO mice (*Figure 3E*), consistent with the findings that GRP-GRPR signaling is not required for histamine itch (*Sun et al., 2009*). Together, these results show that the role of BNP signaling in the spinal cord is dependent on NMB-NMBR signaling and independent of GRP-GRPR signaling.

Next, we evaluated whether NMB has a modulatory function that resembles BNP in histamine itch. Mice pretreated with NMB (0.05 nmol, i.t.) did not exhibit enhanced scratching behaviors evoked by either histamine or CQ (*Figure 3F and G*), ruling out the possibility that NMB would function as a modulator. To determine whether NPRC facilitates histamine itch, we tested the effect of ANP-4–23, a selective NPRC receptor agonist (*Maack et al., 1987*), on NMB-evoked scratching behavior. Although ANP-4–23 or NMB at the low dose could not induce substantial scratching behavior individually, their combined administration (i.t.) evoked robust scratching behavior (*Figure 3H*), demonstrating that ANP-4–23 could potentiate NMB action. To further determine the role of NPRC in itch, we pharmacologically inhibited NPRC with AP 811 (i.t), a highly selective NPRC antagonist (*Koyama et al., 1994*), and found that AP 811 significantly reduced BNP facilitated histamine itch (*Figure 3I*). Given that NMBR acts through the canonical $G_q$-PLC-$Ca^{2+}$ signaling in histamine itch (*Wan et al., 2017*; *Zhao et al., 2014b*), we also tested whether NPRC is coupled to NMBR to facilitate histamine itch. Indeed, BNP-facilitated histamine itch was markedly reduced by U 73122 treatment, a selective PLC inhibitor (*Figure 3I*), suggesting an intracellular coupling between NPRC and NMBR.

We previously showed that NMB exerts its role exclusively through NMBR in the spinal cord, as NMB is a functional antagonist for GRPR in spite of its cross-binding activity with GRPR in the spinal cord (*Zhao et al., 2014b*). Double RNAscope ISH showed that ~ 60% of *Nppb* neurons co-expressed *Nmb* in the DRG (*Figure 3J and K*), whereas *Grp* neurons in DRGs showed minimal co-expression (~19%) with *Nppb* (*Figure 3L and M*). This prompted us to test whether BNP may facilitate NMB/histamine itch signaling through crosstalk between NMBR, which is required for histamine itch, and NPRA or NPRC, two receptors that bind to BNP.

## BNP facilitates NMB-evoked calcium response via NPRC-NMBR crosstalk

To test whether BNP can potentiate NMBR function, we took advantage of the fact that NMB exclusively activates NMBR neurons in the spinal cord (*Zhao et al., 2014b*) and examined the response of NMBR neurons to NMB using $Ca^{2+}$ imaging of dorsal horn neurons (*Munanairi et al., 2018*). NMBR functions via the canonic $G_q$ coupled PLCβ-$Ca^{2+}$ signaling analogous to GRPR (*Liu et al., 2011*; *Zhao et al., 2014a*). Using a protocol for investigating facilitating effect (*Zhao et al., 2014a*), we found that NMB at 20 nM, but not at 10 nM, was able to induce $Ca^{2+}$ transient in perspective NMBR neurons identified with the first application (*Figure 4A and B*). BNP by itself rarely induce $Ca^{2+}$ transient, regardless of the dose, in these neurons. However, when BNP (200 nM) was co-applied with a subthreshold concentration of NMB (10 nM), it dramatically potentiated $Ca^{2+}$ transients in response to the second NMB application (*Figure 4B*). Overall, of 1513 neurons analyzed, 100 responded to 20 nM NMB (6.6 %) and 16 to BNP (200 nM, 1.1 %). Thus were classified as NMBR neurons. From which, 16 responded to co-application of BNP (200 nM, 1.1 %). From 33 NMBR neurons used in this study, 8 (24%) were potentiated by co-application of NMB (10 nM) and BNP. Notably, the percentage of NMBR neurons that responded to both NMB and BNP is largely consistent with the finding that 29% of which

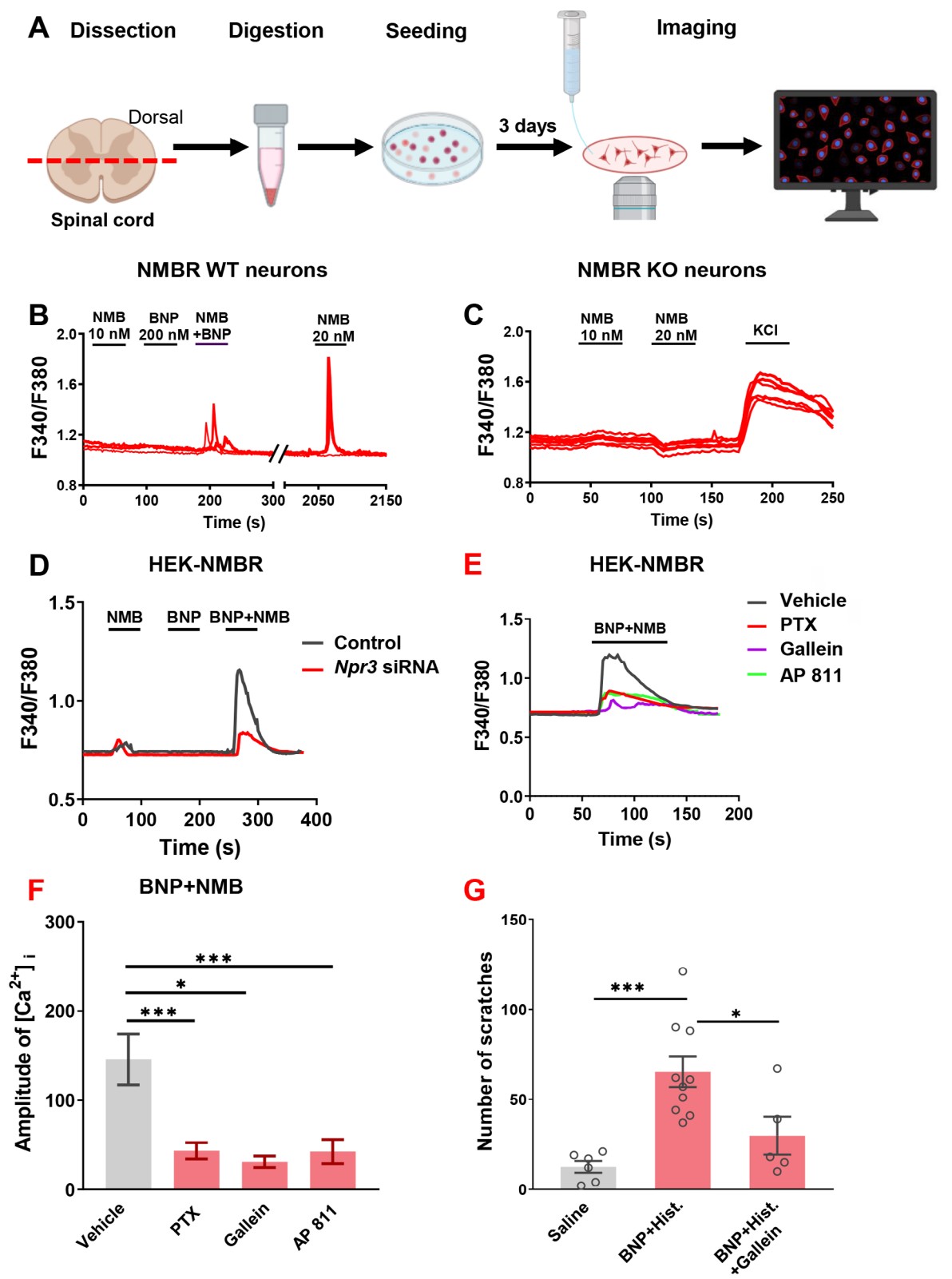

**Figure 4.** Potentiation of NMB-evoked calcium and scratching responses by BNP requires $G_i$-$G_q$ crosstalk between NPRC-NMBR. (**A**) A diagram showing the procedure for calcium imaging on dissociated spinal cord dorsal horn neurons. (**B**) Sample traces showing that co-application of BNP and NMB at low doses evoked $Ca^{2+}$ transients in WT dorsal horn neurons (n = 8 neurons from 33 NMBR neurons analyzed, n = 10 pups). These neurons responded to both BNP/NMB at the low doses responded to NMB at 20 nM robustly, indicating that they are healthy neurons. (**C**) No dorsal horn neurons responded

*Figure 4 continued on next page*

*Figure 4 continued*

to NMB (20 nM) isolated from the spinal cord of *Nmbr* KO mice (n = 2 mice), whereas they responded to KCl, indicating that they were healthy neurons. (**D**) Co-application of BNP (1 µM) with subthreshold of NMB (1 pM) evoked robust calcium response in HEK 293 cells co-expressing NMBR, which was significantly attenuated by *Npr3* siRNA treatment. (**E**) Calcium transients induced by BNP and NMB were attenuated by pre-incubation of PTX (200 ng/ml), gallein or AP 811 (0.1 µM) for 30 min. n = 6 slides per group with at least 50 cells imaged on each slide. (**F**) Quantification of calcium concentration ([Ca2+]i) of **E**. (**G**) I.t. gallein (20 nmol) significantly reduced scratching behavior evoked by histamine (25 µg, i.d.) facilitated with BNP (30 µM, i.t.). Values are presented as mean ± SEM, n = 6–10. *p < 0.05, ***p < 0.001, one-way ANOVA followed by Tukey's test.

The online version of this article includes the following source data and figure supplement(s) for figure 4:

**Source data 1.** Potentiation of NMB-evoked calcium and scratching responses by BNP requires $G_i$-$G_q$ crosstalk between NPRC-NMBR.

**Figure supplement 1.** real-time RT-PCR detected endogenous expression of *Npr1*, *Npr2*, and *Npr3* in HEK 293 cells.

express *Npr3*. To further evaluate the response specificity of NMBR neurons to NMB, we analyzed the response of the dorsal horn neurons lacking *Nmbr* to NMB (up to 20 nM). Importantly, none of NMBR KO neurons showed $Ca^{2+}$ transients (n = 196 neurons), albeit they all responded to KCL (*Figure 4C*). Together, these results confirmed the identity as well as response specificity of NMBR neurons to NMB (*Figure 4C*).

To probe the possibility of NPRC-NMBR crosstalk, we took advantage of the fact that HEK 293 cells express endogenous *Npr1* and *Npr3* as shown by qRT-PCR (*Figure 4—figure supplement 1*). In HEK 293 cells stably expressing NMBR, application of neither BNP (1 µM) nor NMB (1 pM) induced $Ca^{2+}$ response, whereas their co-application evoked robust $Ca^{2+}$ transients (*Figure 4D*). Importantly, the effect of BNP was greatly attenuated by *Npr3* siRNA treatment (*Figure 4D*), indicating that BNP facilitates NMB/NMBR signaling through NPRC. NPRC has been linked to the inhibition of adenylate cyclase (AC)/cAMP signaling and can activate pertussis toxin (PTX)-sensitive $G_{\alpha i/\beta\gamma}$ signaling pathway (*Anand-Srivastava et al., 1990*). To examine whether the $G_{\alpha i/\beta\gamma}$ pathway is involved in the facilitatory effect of BNP/NPRC, HEK 293 cells were treated with pertussis toxin (PTX) to inactivate $G_{\alpha i}$ protein (*Murayama and Ui, 1983*). Subsequent incubation of BNP and NMB induced much smaller $Ca^{2+}$ spikes compared to control cells (*Figure 4E*). The amplitude of intracellular $Ca^{2+}$ concentrations ([$Ca^{2+}]_i$) was significantly reduced by PTX treatment (*Figure 4F*). Pre-incubation of gallein, a small molecule $G_{\beta\gamma}$ inhibitor, also blocked the facilitation effect of BNP on NMB-induced calcium spikes (*Figure 4E and F*). Consistently, i.t. gallein markedly attenuated the facilitatory effect of BNP on histamine itch (*Figure 4G*). Collectively, these results indicate that BNP-NPRC signaling potentiates NMB/NMBR signaling via $G_{\alpha i/\beta\gamma}$ signaling.

## NPRA and NPRC neurons are required for histamine itch

BNP-saporin (BNP-sap) has been used to ablate NPRA neurons in the spinal cord (*Mishra and Hoon, 2013*). Nevertheless, the expression of NPRC in the dorsal horn has raised the question of whether BNP-sap may additionally ablate neurons expressing NPRC (*Figure 1H–N*). Because BNP-sap at 5 µg, as described previously (*Mishra and Hoon, 2013*), resulted in the lethality of WT mice, we reduced the dose to 2.5 µg so that enough animals could survive for behavioral and molecular analysis. RNAscope ISH showed that the number of $Npr1^+$ + were reduced to ~50% (82.9 ± 3.3 in control vs. 39.8 ± 2.2 in BNP-sap) (*Figure 5A,F*), whereas $Npr2^+$ neurons were not affected (288.0 ± 18.2 in control vs. 300.5 ± 7.8 in BNP-sap) (*Figure 5B and F*). Moreover, BNP-sap ablated ~67% of $Npr3^+$ neurons (41.8 ± 1.4 in control vs. 13.8 ± 1.8 in BNP-sap) as well as ~37% of $Grp^+$ + (44.6 ± 1.9 in control vs. 22.6 ± 2.0 in BNP-sap) (*Figure 5C, D and F*). As expected, the number of $Nmbr^+$ neurons was also significantly reduced after BNP-sap injection, likely due to *Npr3* expression in these neurons (*Figure 5E and F*). A premise for ablation of neurons with peptide-conjugated saporin approach is the internalization of the receptor upon binding to the saporin, resulting in cell death (*Wiley and Lappi, 2003*). To test whether BNP can also internalize NPRB and NPRC, HEK 293 cells were transfected with *Npr1, 2*, and *3* cDNA tagged with mCherry (mCh) separately. Indeed, BNP internalized NPRA and NPRC, but not NPRB, in HEK 293 cells (*Figure 5G*), indicating that BNP-sap could ablate both NPRA and NPRC cells. Interestingly, behavioral studies showed that histamine itch was significantly reduced in BNP-sap mice (*Figure 5H*), whereas CQ itch was not affected (*Figure 5I*). These results suggest that NPRA and NPRC neurons in the spinal cord play an important role in histamine itch, which can be attributed to partial ablation of NMBR neurons.

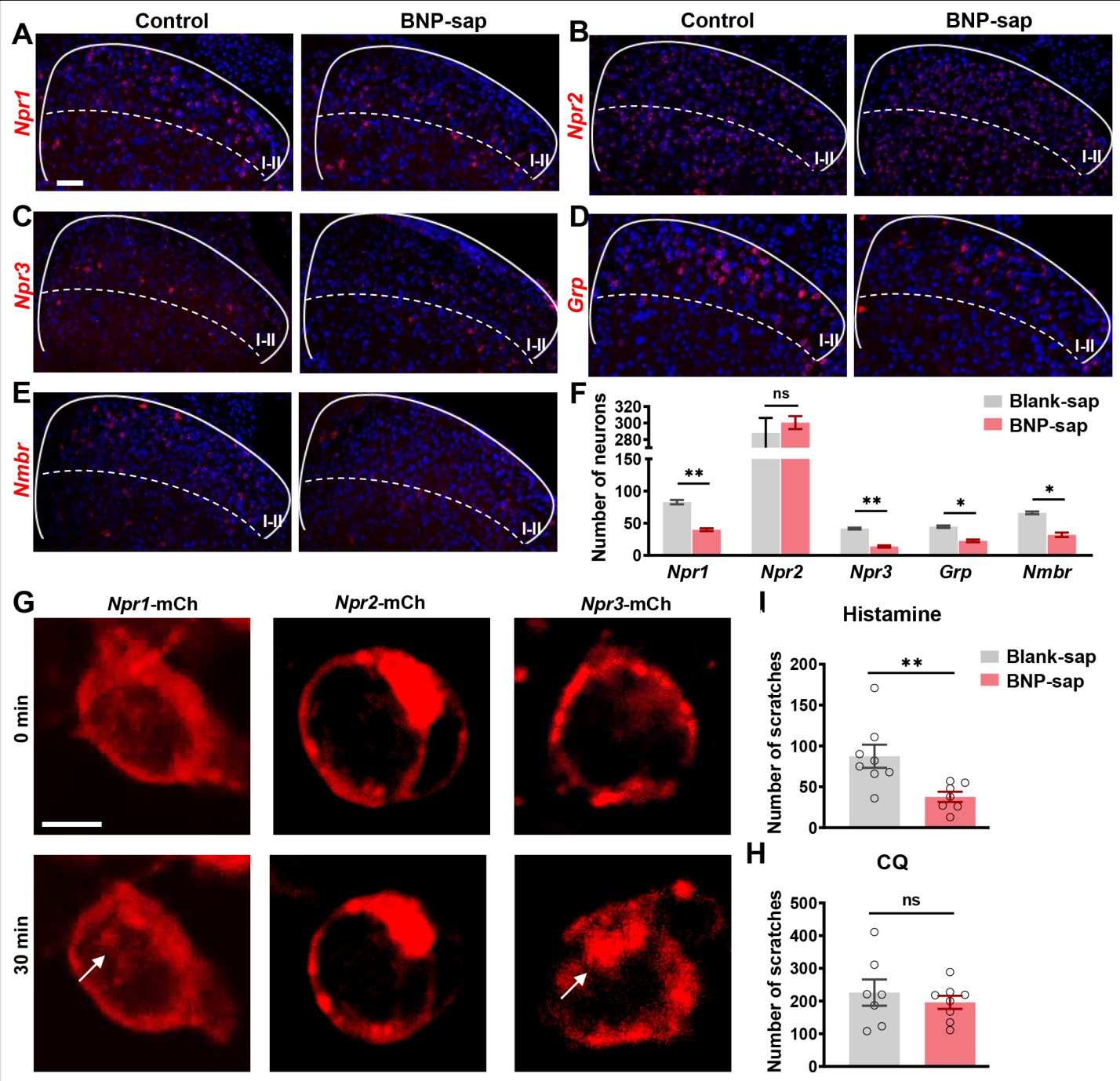

**Figure 5.** BNP-sap ablates spinal cord neurons expressing *Npr1* and *Npr3*. (**A-F**) RNAscope ISH images (**A and C**) and quantified data (**F**) showing that BNP-sap ablated *Npr1+* (**A**), *Npr3+* (**C**), *Grp+* (**D**), and *Nmbr+* (**E**) neurons (red) in the dorsal horn of the spinal cord, while *Npr2+* (**B**) neurons (red) were not affected. n = 4. (**G**) Incubation of BNP (10 μM) for 30 min caused internalization of *Npr1*-mCh and *Npr3*-mCh in HEK 293 cells transfected with NMBR cDNA as indicated by arrows. No internalization of *Npr2*-mCh was observed. Scale bar, 20 μm. mCh: mCherry. (**H, I**) Scratching behaviors induced by histamine (**H**), but not CQ (**I**) were significantly reduced in BNP-sap treated mice. n = 7–8. Values are presented as mean ± SEM. *p < 0.05, **p < 0.01, unpaired t test. Scale bar, 50 μm in **A–F**, 10 μm in **G**.

The online version of this article includes the following source data for figure 5:

**Source data 1.** BNP-sap ablates spinal cord neurons expressing *Npr1* and *Npr3*.

## The function of BNP and NPRA in dry skin and neuropathic itch

To explore whether NPRA is important for the development of chronic itch mediated by nonhistaminergic mechanisms, we used the mouse dry skin model deprived of histaminergic component (*Miyamoto et al., 2002*). From day 8, *Npr1* KO mice appeared to show a tendency toward reduced scratching behavior relative to their WT littermates. The differences, however, were not statistically significant (*Figure 6—figure supplement 1* A). The normal dry skin itch in *Npr1* KO mice prompted us to examine *Nppb* expression in DRGs of dry skin mice. Real-time RT-PCR results showed that the levels of *Nppb* mRNA in dry-skin mice were either significantly reduced or not changed depending on the day examined (*Figure 6—figure supplement 1B*, data not shown). Similarly, dry-skin mice also showed unchanged expression of somatostatin (*Sst*), a peptide largely co-localized with *Nppb* in DRG neurons (*Huang et al., 2018*; *Stantcheva et al., 2016*) and *Tac1* (*Figure 6—figure supplement 1B-D*). In contrast, *Grp* and *Nmb* expression levels in mice with dry skin were increased by 447% and 87%, respectively (*Figure 6—figure supplement 1B*). These findings suggest that BNP-NPRA signaling is not required for the development of dry skin itch.

SST type two receptor (SST2R) is expressed in GABAergic neurons in the spinal cord and has been considered to be a sole receptor for SST (*Kardon et al., 2014*; *Polgár et al., 2013*). To test whether SST2R neurons may inhibit both itch and pain transmission, we pharmacologically inhibit these neurons by i.t. SST injection followed by validating the nature of evoked scratching behavior since the injection onto the nape may induce itch-, pain-related or undefined scratching behavior (*Shimada and LaMotte, 2008*). I.t. SST-evoked scratching was markedly reduced but not abolished by intraperitoneal injection (i.p.) of morphine (*Figure 6A*), a method used to evaluate whether i.t. induced scratching/biting behavior reflects pain (*Hylden and Wilcox, 1981*). In addition, scratching behaviors evoked by i.t. injection of SST and octreotide (OCT), a selective SST2R agonist, were significantly attenuated on mice with bombesin-saporin (BB-sap) treatment, which can completely block nonhistaminergic itch transmission (*Sun et al., 2009*; *Figure 6B and C*). These results suggest that the pharmacological inhibition of SST2R neurons could result in disinhibition of both itch and pain transmission.

We then evaluated the expression of *Nppb* and *Sst* in neuropathic itch using BRAF$^{Nav1.8}$ mice that developed spontaneous scratching behavior resulting from enhanced expression of itch-sensing peptides/receptors in sensory neurons (*Zhao et al., 2013*). Interestingly, *Nppb* was dramatically down-regulated, whereas *Sst* was barely detectable in DRG neurons of BRAF$^{Nav1.8}$ mice (*Figure 6—figure supplement 1C, D, E*). The reduced *Sst* expression may reflect the dampening effect of the dorsal horn GABAergic neuronal activity under neuropathic itch conditions. Thus, BNP and SST are not involved in the development of dry skin and neuropathic itch in mice.

If a subset of primary afferents exclusively expresses itch- but little pain-related neuropeptides, it can be predicted that cutaneous activation of these afferents would evoke itch-related scratching behavior. For example, optical activation of the skin innervated by *Grp* primary afferents evoked frequency-dependent itch-related scratching behavior (*Barry et al., 2020*). To further explore whether BNP-expressing afferents are itch-specific, we used *Sst*-Cre mice as a surrogate to perform optical stimulation of skin-innervating *Sst*-expressing fibers. *Sst*-Cre mice were crossed with Ai32 mice to generate Sst-ChR2 mice that express channelrhodopsin-2/EYFP or ChR2/EYFP in the Sst locus, as confirmed by highly overlapping expression of YFP and *Sst* (*Figure 6F*). Sst-ChR2 or Sst-cre mice were stimulated with 473 nm blue light with a fiber optic held just above the nape skin (15 mW power from fiber tip) at 1, 5, 10, or 20 Hz with a 3 s On-Off cycle for 5 minutes (*Figure 6D* and *Figure 6—video 1*). Stimulation at all frequencies failed to evoke significant scratching behaviors in Sst-ChR2 mice compared to *Sst-cre* mice (*Figure 6E*). At last, we analyzed the expression of Sst-ChR2 in DRGs. Consistent with previous studies (*Stantcheva et al., 2016*), we found that Sst-ChR2 sensory neurons do not co-express the peptidergic marker calcitonin gene-related peptide (CGRP), nor do they show the non-peptidergic Isolectin B4 (IB4)-binding (*Figure 6G*). However, some Sst-ChR2 sensory neurons do co-express the myelinated marker neurofilament heavy (NF-H) and transient receptor potential cation channel subfamily V member 1 (TRPV1) (*Figure 6G*). Examination of the hairy nape skin revealed that expression of Sst-ChR2 in the epidermis as well as expression in some hair follicles of the dermis within apparent lanceolate endings (*Figure 6H*).

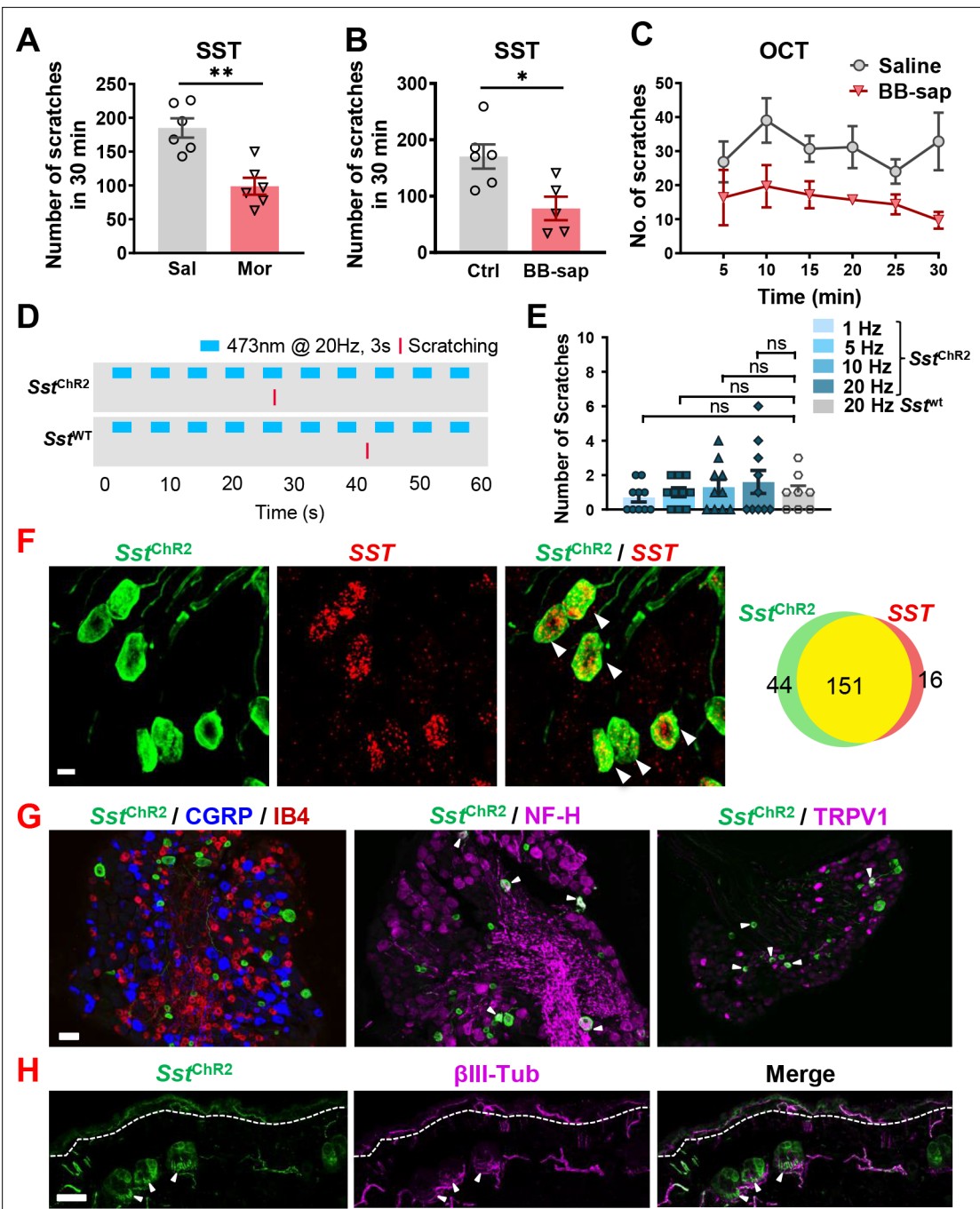

**Figure 6.** SST evoked both pain and itch responses in mice. (**A**) Pre-injection of morphine (10 mg/kg, i.p.) for 30 min attenuated scratching behaviors induced by i.t. injection of SST (5 nmol). n = 6 mice per group. Sal, saline; Mor, morphine. (**B, C**) SST (5 nmol, i.t.)(**B**) and OCT (**C**) -evoked scratching behaviors were significantly reduced in bombesin-saporin-treated mice comparing with control mice that were treated with blank saporin. n = 5–6 mice per group. Ctrl, control; BB-sap, bombesin-saporin. (**D**) Raster plot of scratching behavior induced by light stimulation of skin in Sst-ChR2 and *Sst-cre* mice. (**E**) Number of scratches in 5 min induced by 3 s – 1, 5, 10, or 20 Hz light stimulation of nape skin in Sst-ChR2 and *Sst-cre* mice. n = 8–10 mice. ns – not significant, one-way ANOVA with Tukey *post hoc*. (**F**) IHC images of Sst-ChR2/*Sst* co-expression in DRG of Sst-ChR2 mice (Left). Arrowheads indicate co-expression. Scale bar, 10 µm. Venn diagram showing overlapping expression of Sst-ChR2 and *Sst* in DRG neurons (Right). (**G**) IHC images of Sst-ChR2/CGRP/IB4 (left), Sst-ChR2/NF-H (middle), and Sst-ChR2/TRPV1 (right) in DRG of Sst-ChR2 mice. Arrowheads indicate co-expression. (**H**) IHC image of Sst-ChR2/βIII-Tubulin in hairy nape skin. Dashed line marks epidermal/dermal boundary. Arrowheads indicate ChR2 expression in lanceolate endings of hair follicles. Values are presented as mean ± SEM. *p < 0.05, **p < 0.01, unpaired t test. Scale bars, 10 µm in **F**, 100 µm in **G** and **H**.

The online version of this article includes the following video, source data, and figure supplement(s) for figure 6:

*Figure 6 continued on next page*

*Figure 6 continued*

**Source data 1.** SST evoked both pain and itch responses in mice.

**Figure supplement 1.** BNP-NPRA signaling is dispensable for nonhistaminergic itch and neuropathic itch.

**Figure supplement 1—source data 1.** BNP-NPRA signaling is dispensable for histamine-independent chronic itch.

**Figure 6—video 1.** Optogenetic stimulation of skin of Sst-ChR2 mice failed to induce scratching behaviors.

https://elifesciences.org/articles/71689/figures#fig6video1

## Discussion

In this study, we demonstrate that in sensory neurons, BNP can function as a neuromodulator, rather than a neurotransmitter, to facilitate itch transmission. We also show that NPRC in the spinal cord is crucial for mediating BNP-facilitated histaminergic itch. $Ca^{2+}$ signaling is a hallmark feature of neuronal activation (*Berridge, 1998*). By demonstrating that BNP alone could not activate $Ca^{2+}$ transients either in HEK293 cells or in the dorsal horn neurons, we verify that BNP is an inhibitory neuropeptide, in line with the observation that i.t. BNP, even at very high doses (2.5–5 μg), fails to induce the rapid onset of scratching behavior, typical for ligand-mediated acute activation of excitatory itch receptors in the spinal cord (*Sun and Chen, 2007*; *Wan et al., 2017*; *Zhao et al., 2014b*). Such high doses therefore most likely fall outside the range of its endogenous concentration (e.g. at picomolar concentrations) required for mediating acute itch transmission directly (*Wan et al., 2017*). Although *Npr1* siRNA knockdown in both spinal cord and DRGs makes it difficult to ascribe itch deficits to either region, the observation that BNP-sap treatment, in spite of partial ablation, attenuates histamine but not CQ itch suggests that NPRA in DRGs, rather than in the spinal cord, is involved in CQ itch. Whether NPRA may facilitate histaminergic itch in DRGs awaits further clarification. Given a very small fraction of spinal NPRA neurons also express NMBR, the possibility that BNP-NPRA signaling may marginally contribute to histaminergic itch cannot be excluded. Coupled with previous studies suggesting the inhibitory effect of BNP on nociceptive neurons (*Liu et al., 2014*; *Zhang et al., 2010*), it remains plausible that NPRA in DRGs may exert opposing functions by facilitating itch while inhibiting certain types of inflammatory pain. Nonetheless, the observation that most *Nppb* neurons express little GRP implies that BNP/NMB and GRP released from distinct types of primary afferents may be responsible for the activation of the dorsal horn neurons expressing NPRC/NMBR and GRPR, respectively.

Perhaps, the most unexpected finding is that the BNP-NPRC signaling facilitates histamine itch through NPRC-NMBR crosstalk. Given that PTX acts on $G_{\alpha i}$ directly in the $G_{\alpha\beta\gamma}$ heterotrimer (*Smrcka, 2008*) and gallein acts on $G_{\beta\gamma}$ directly, the findings that inhibition of $G_{\alpha i}$ and $G_{\beta\gamma}$ signaling similarly attenuates the facilitatory effect of BNP on NMB-induced $Ca^{2+}$ response in NPRC/NMBR cells and BNP-facilitated histamine itch suggest an intracellular coupling of the $G_{\alpha i/\beta\gamma}$ signaling pathway to PLCβ signaling downstream of NMBR (*Figure 7A*). Numerous studies have shown that inhibitory $G_i$-coupled receptors enhance or facilitate the activation of the canonical PLCβ–$Ca^{2+}$ signaling transduction pathway downstream of $G_q$-coupled receptors (*Prezeau et al., 2010*; *Werry et al., 2003*). For example, we have shown that inhibitory receptors such as 5-HT1A and μ-opioid receptor isoform MOR1D in mice or MOR1Y in humans could facilitate or activate $Ca^{2+}$ signaling downstream of GRPR via intracellular $G_i$-$G_q$ crosstalk (*Liu et al., 2019*; *Liu et al., 2011*; *Zhao et al., 2014a*). Nevertheless, NPRC-NMBR represents the first example of the crosstalk between a non-GPCR inhibitory receptor and an excitatory GPCR. The attenuation of NPRC-GRPR crosstalk by pharmacological inhibition of either $G_{\beta\gamma}$ or PLCβ signaling suggests that, irrespective of the receptor type, $G_i$-$G_q$ coupling leading to $Ca^{2+}$ mobilization are versatile and universal mechanisms. In congruence with this, even the rare $G_i$-$G_{\beta\gamma}$-PLCβ-$Ca^{2+}$ signaling pathway that was considered to be a stand-alone paradigm was recently found to be dependent on $G_q$ signaling (*Pfeil et al., 2020*). On the other hand, partial attenuation of BNP-facilitated histaminergic itch by PTX and gallein implies that $G_{\beta\gamma}$-independent signaling mechanisms may also have a role in NPRC-NMBR crosstalk. It is intriguing that ANP may not play a role in facilitating histamine itch, even though it can bind to NPRC with equal potency as BNP. It is possible that ANP may interact with the receptor at different binding sites (*Savoie et al., 1995*). Alternatively, it could be due to a much faster clearance rate of ANP (0.5–4 min) than BNP (four to more than 20 min) or CNP (*Potter, 2011*).

Our study clarifies the role of SST and *Nppb/Sst* fibers in itch. The finding that cutaneous activation of BNP/SST fibers failed to evoke scratching behavior suggests that activation of these fibers does

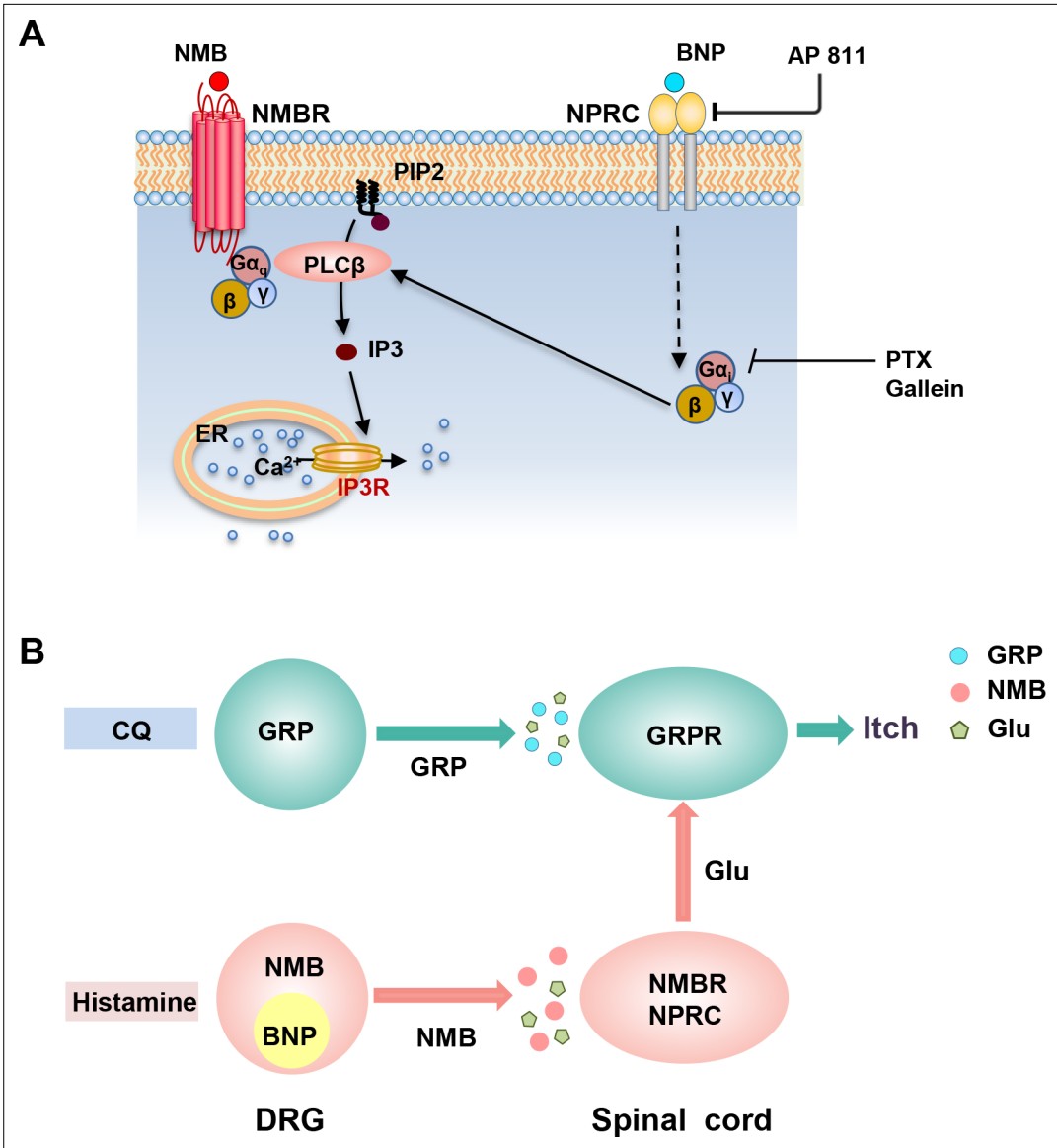

**Figure 7.** Schematics for the BNP-NPRC facilitated signaling pathway and distinct neuropeptide pathways for histamine-dependent and -independent itch. (**A**) A schematic showing a model for NMBR-NPRC cross-signaling facilitated by BNP via the NMB-NMBR pathway. In response to histamine, NMB and BNP are released from primary afferents to activate NMBR and NPRC concurrently. Activation of NMBR by NMB at a low concentration may prime PLCβ signaling, whereas activation of NPRC by BNP stimulates Gai signaling, which in turn stimulates PLCβ to activate downstream Ca2+ signaling. (**B**) A hypothetic model depicting the respective roles of neuropeptides and glutamate in itch transmission. CQ itch is mediated in part by GRP-GRPR signaling independent of glutamatergic transmission. In contrast, histamine itch is mediated by NMB-NMBR signaling from primary afferents to NMBR neurons and by glutamatergic transmission from NMBR neurons to GRPR neurons. BNP facilitates NMB-NMBR signaling via NPRC independent of GRP-GRPR signaling but dependent on GRPR neurons. Glu: glutamate; GRP: gastrin-releasing peptide; BNP: B-type natriuretic peptide; NMB: neuromedin B.

The online version of this article includes the following figure supplement(s) for figure 7:

**Figure supplement 1.** A hypothetic model depicting the role of BNP, NMB, and SST in facilitation of itch and disinhibition of pain, respectively.

not transmit itch information as a functional entity. These results may seem surprising in light of robust scratching behavior evoked by i.t. SST. However, they are consistent with the observation that DRG-specific deletion of *Sst* did not influence itch transmission (*Huang et al., 2018*). Although diminished itch behavior of Wnt1Cre/*Sst*f/f mice was used to argue for a role of SST in itch disinhibition in sensory

neurons (*Huang et al., 2018*), the deficit can be ascribed to deletion of *Sst* in the brain regions, where Wnt1 is expressed in numerous neural precursors (*Lewis et al., 2013*). It is possible that cutaneous activation of *Nppb/Sst* fibers may provoke the release of BNP/SST onto the spinal cord; However, their endogenous release may not be sufficient to evoke scratching behavior. Together, these loss- and gain-of-function studies indicate that itch-related scratching behavior evoked by i.t. SST represents a pharmacological artifact. Interestingly, conditional knockout of *Sst* in DRGs attenuated thermal and mechanical pain (*Huang et al., 2018*), suggesting that SST release may contribute to certain types of nociceptive transmission. Consistently, the downregulation of *Sst* in mice with chronic itch may reflect a dampening effect of spinal inhibitory neuronal activity for nociceptive transmission under pathological itch conditions. Thus, the role of SST in sensory neurons is limited to the disinhibition of certain types of pain by inhibiting SST2R inhibitory neurons gating nociceptive transmission (*Figure 7—figure supplement 1*). This interpretation is in line with recent studies suggesting that spinal inhibitory neurons that gate itch and pain, respectively, are anatomically segregated (*Chen, 2021*).

Most human chronic itch conditions are resistant to antihistamines, which is a major focus of current research due to clinical implications. However, current mouse models of chronic itch, including allergic contact dermatitis (ACD) and atopic dermatitis (AD), are additionally mediated by histamine-dependent mechanisms, as they are chemically induced allergic inflammatory responses that involve histamine release from mast cells (*Shim and Oh, 2008*; *Wang and Kim, 2020*). In these mouse models, BNP may be upregulated in DRGs as a result of mast cell degranulation and may facilitate histaminergic itch (*Liu et al., 2020*; *Solinski et al., 2019*). However, it is either downregulated or not altered in a dry skin itch model, which is mediated exclusively through GRP-dependent nonhistaminergic mechanism (*Akiyama et al., 2010*; *Miyamoto et al., 2002*; *Zhao et al., 2013*). Consistently, NPRA is dispensable for the development of dry skin itch and *Nppb* is downregulated in a neuropathic itch model as shown in the present study.

In summary, we demonstrate a novel BNP-NPRC-NMBR crosstalk in the modulation of itch transmission (*Figure 7A*). Considering that BNP potently enhances NMB function, but not vice versa, these studies suggest that neuropeptides in sensory neurons either encode or modulate itch information (*Figure 7B*). Our studies reveal an unexpected role of NPRC in facilitating NMB-mediated histaminergic itch transmission (Figure S4). These results delineate distinct functions of GRP, NMB, and BNP in histaminergic and nonhistaminergic itch and highlight the different modes of action for neuropeptides in the coding and modulating of itch (*Figure 7B*; *Chen, 2021*).

## Materials and methods

**Key resources table**

| Reagent type (species) or resource | Designation | Source or reference | Identifiers | Additional information |
|---|---|---|---|---|
| Strain, strain background (*Mus musculus*) | C57Bl/6 mice | The Jackson Laboratory | Cat#000664 | NA |
| Strain, strain background (*Mus musculus*) | *Npr1* KO | The Jackson Laboratory | Cat#004374 | NA |
| Strain, strain background (*Mus musculus*) | *Grpr* KO mice | The Jackson Laboratory | Cat#003126 | NA |
| Strain, strain background (*Mus musculus*) | *Nmbr* KO | *Ohki-Hamazaki et al., 1999* | NA | NA |
| Strain, strain background (*Mus musculus*) | Ai32 (*Gt(ROSA)26Sor^{tm32(CAG-OP4*H134R/EYFP)Hze}*) | The Jackson Laboratory | Cat#024109 | NA |
| Strain, strain background (*Mus musculus*) | *Sst^{Cre}* | The Jackson Laboratory | Cat#018973 | NA |

*Continued on next page*

*Continued*

| Reagent type (species) or resource | Designation | Source or reference | Identifiers | Additional information |
|---|---|---|---|---|
| Cell line (human) | HEK 293 | ATCC | Cat#CRL-1573 | NA |
| Antibody | Rabbit anti-CGRPα (Rabbit polyclonal) | Millipore | Cat#AB1971 | IF (1/3000) |
| Antibody | Guinea pig anti-Substance P (Guinea pig polyclonal) | Abcam | Cat#ab10353 | IF (1/1000) |
| Antibody | Guinea pig anti-TRPV1 (Guinea pig polyclonal) | Neuromics | Cat#GP14100 | IF (1/1000) |
| Antibody | Chicken anti-NF-H (Chicken polyclonal) | EnCor Biotechnology | Cat#CPCA-NF-H | IF (1/2000) |
| Antibody | Rabbit anti-βIII-Tubulin (Rabbit polyclonal) | Biolegend | Cat#802,001 | IF (1/2000) |
| Antibody | Rabbit anti-GFP (Rabbit polyclonal) | Molecular Probes | Cat#A11122 | IF (1/1000) |
| Antibody | Chicken anti-GFP (Chicken polyclonal) | Aves Labs | Cat#GFP-1020 | IF (1/500) |
| Antibody | FITC-conjugated Isolectin B4 (Polyclonal) | Sigma | Cat#L2895 | IF (1/500) |
| Antibody | IB4-AlexaFluor 568 conjugate (Polyclonal) | ThermoFisher Scientific | Cat#I21412 | IF (1/500) |
| Antibody | Cy3 conjugated donkey anti-mouse IgG (Polyclonal) | Jackson ImmunoResearch | Cat#715-165-150 | IF (1/500) |
| Antibody | Cy3 conjugated donkey anti-chicken IgG (Polyclonal) | Jackson ImmunoResearch | Cat#703-165-155 | IF (1/500) |
| Antibody | Cy3 conjugated donkey anti-rabbit IgG (Polyclonal) | Jackson ImmunoResearch | Cat#711-165-152 | IF (1/500) |
| Antibody | Cy3 conjugated donkey anti-guinea pig IgG (Polyclonal) | Jackson ImmunoResearch | Cat#706-165-148 | IF (1/500) |
| Antibody | Cy5 conjugated donkey anti-mouse pig IgG (Polyclonal) | Jackson ImmunoResearch | Cat#715-175-150 | IF (1/500) |
| Antibody | Cy5 conjugated donkey anti-chicken IgG (Polyclonal) | Jackson ImmunoResearch | Cat#703-175-155 | IF (1/500) |
| Antibody | Cy5 conjugated donkey anti-rabbit IgG (Polyclonal) | Jackson ImmunoResearch | Cat#711-175-152 | IF (1/500) |
| Antibody | Cy5 conjugated donkey anti-guinea pig IgG (Polyclonal) | Jackson ImmunoResearch | Cat#706-175-148 | IF (1/500) |
| Antibody | FITC conjugated donkey anti-mouse IgG (Polyclonal) | Jackson ImmunoResearch | Cat# 715-095-150 | IF (1/500) |
| Antibody | FITC conjugated donkey anti-chicken IgG (Polyclonal) | Jackson ImmunoResearch | Cat#703-095-155 | IF (1/500) |
| Antibody | FITC conjugated (Polyclonal)donkey anti-rabbit IgG | Jackson ImmunoResearch | Cat#111-095-144 | IF (1/500) |
| Antibody | FITC conjugated donkey anti-guinea pig IgG (Polyclonal) | Jackson ImmunoResearch | Cat#706-095-148 | IF (1/500) |
| Peptide, recombinant protein | ANP | GenScript | Cat#RP11927 | 5–10 µg, i.t. |
| Peptide, recombinant protein | BNP | GenScript | Cat#RP11119 | 1–5 µg, i.t. |
| Peptide, recombinant protein | CNP | GenScript | Cat#RP11110 | NA |

*Continued on next page*

*Continued*

| Reagent type (species) or resource | Designation | Source or reference | Identifiers | Additional information |
|---|---|---|---|---|
| Peptide, recombinant protein | SST | GenScript | Cat#RP10230 | 5 nmol, i.t. |
| Peptide, recombinant protein | OCT | GenScript | Cat#SMS 201–995 | NA |
| Peptide, recombinant protein | GRP18-27 | Bachem | Cat#H-3120.0005 | NA |
| Peptide, recombinant protein | NMB | Bachem | Cat#H-3280.0001 | 0.5 nmol, i.t. |
| Chemical compound, drug | Histamine | Sigma | Cat#H7250 | 100 µg, i.d. |
| Chemical compound, drug | Chloroquine | Sigma | Cat#C6628 | 200 µg, i.d. |
| Chemical compound, drug | BNP-saporin (BNP-sap) | Advanced Targeting Systems | Cat#IT-69 | 2.5 µg/mouse, i.t |
| Chemical compound, drug | Blank-saporin | Advanced Targeting Systems | Cat#IT-27B | NA |
| Chemical compound, drug | Pertussis toxin (PTX) | R&D Systems | Cat#3,097 | 200 ng/ml |
| Chemical compound, drug | Gallein | R&D Systems | Cat#3,090 | 100 µM, 2 mM calcium imaing |
| Chemical compound, drug | Acetone | Sigma | Cat#179,124 | NA |
| Chemical compound, drug | AP 811 | Tocris | Cat#5,498 | 10 µM, i.t. |
| Chemical compound, drug | ANP 4–23 | Bachem | Cat#4030384 | 10 µg, i.t. |
| Chemical compound, drug | U73122 | Selleck | Cat#S8011 | 13.5 nmol, i.t. |
| Chemical compound, drug | Gallein | Selleck | S5978 | 20 nmol, i.t., behavior study |
| Chemical compound, drug | Ddiethyl ether | Sigma | Cat#309,966 | NA |
| Sequence-based reagents | RNAscope Fluorescent Multiplex Assay v2 | Advanced Cell Diagnostics | Cat#323,110 | NA |
| Sequence-based reagents | RNAscope probe Mm_*Nppb* | Advanced Cell Diagnostics | Cat#425,021 | NA |
| Sequence-based reagents | RNAscope probe Mm_*Npr1* | Advanced Cell Diagnostics | Cat#484,531 | NA |
| Sequence-based reagents | RNAscope probe Mm_*Npr2* | Advanced Cell Diagnostics | Cat#315,951 | NA |
| Sequence-based reagents | RNAscope probe Mm_*Npr3* | Advanced Cell Diagnostics | Cat#502,991 | NA |
| Sequence-based reagents | RNAscope probe Mm_*Grp* | Advanced Cell Diagnostics | Cat#317,861 | NA |
| Sequence-based reagents | RNAscope probe Mm_*Grpr* | Advanced Cell Diagnostics | Cat#317,871 | NA |
| Sequence-based reagents | RNAscope probe Mm_*Nmbr* | Advanced Cell Diagnostics | Cat#406,461 | NA |

*Continued*

| Reagent type (species) or resource | Designation | Source or reference | Identifiers | Additional information |
|---|---|---|---|---|
| Sequence-based reagents | RNAscope probe Mm_*Vgat* | Advanced Cell Diagnostics | Cat#319,191 | NA |
| Sequence-based reagents | RNAscope probe Mm_*Vglut2* | Advanced Cell Diagnostics | Cat#319,171 | NA |
| Sequence-based reagents | *Npr1* siRNA | Sigma | Cat#SASI_Mm01_00106966 | 2 µg/µL, i.t. |
| Sequence-based reagents | *Npr2* siRNA | Sigma | Cat#SASI_Mm01_00201357 | 2 µg/µL, i.t. |
| Sequence-based reagents | *Npr3* siRNA | Sigma | Cat#SASI_Mm01_00036567 | 2 µg/µL, i.t. |
| Sequence-based reagents | *Nppb* primer for RT-PCR:<br>5'- GTCAGTCGTTTGGGCTGTAAC-3',<br>5'- AGACCCAGGCAGAGTCAGAA-3' | IDT | PCR primers | NA |
| Sequence-based reagents | *Sst* primer for RT-PCR:<br>5'- CCCAGACTCCGTCAGTTTCT –3',<br>5'- CAGCAGCTCTGCCAAGAAGT –3' | IDT | PCR primers | NA |
| Sequence-based reagents | *Npr1* primer for RT-PCR:<br>5'- TGGAGACACAGTCAACACAGC-3',<br>5'- CGAAGACAAGTGGATCCTGAG-3' | IDT | PCR primers | NA |
| Sequence-based reagents | *Npr2* primer for RT-PCR:<br>5'- TGAGCAAGCCACCCACTT-3',<br>5'- AGGGGGCCGCAGATATAC-3' | IDT | PCR primers | NA |
| Sequence-based reagents | *Npr3* primer for RT-PCR:<br>5'- TGCACACGTCTGCCTACAAT-3',<br>5'- GCACCGCCAACATGATTCTC –3' | IDT | PCR primers | NA |
| Sequence-based reagents | *Grpr* primer for RT-PCR:<br>5'-TGATTCAGAGTGCCTACAATCTTC-3',<br>5'-TTCCGGGATTCGATCTG-3' | IDT | PCR primers | NA |
| Sequence-based reagents | *Nmbr* primer for RT-PCR:<br>5'- GGGGGTTTCTGTGTTCACTC –3',<br>5'- CATGGGGTTCACGATAGCTC –3' | IDT | PCR primers | NA |
| Sequence-based reagents | *Actb* primer for RT-PCR:<br>5'-TGTTACCAACTGGGACGACA-3',<br>5'-GGGGTGTTGAAGGTCTCAAA-3' | IDT | PCR primers | NA |
| Sequence-based reagents | *Gapdh* primer for RT-PCR:<br>5'-CCCAGCAAGGACACTGAGCAA-3',<br>5'-TTATGGGGGTCTGGGATGGAAA-3' | IDT | PCR primers | NA |
| Software and algorithms | Prism 6 | GraphPad Software | https://www.graphpad.com/ | NA |
| Software and algorithms | ImageJ | NIH | https://imagej.nih.gov/ij | NA |
| Software and algorithms | Nikon Elements Software | Nikon | https://www.microscope.healthcare.nikon.com/products/software/nis-elements | NA |

## Animals

Male mice between 7 and 12 weeks of age were used for behavioral experiments. C57Bl/6 mice were purchased from The Jackson Laboratory (http://jaxmice.jax.org/strain/013636.html). *Npr1* KO mice (*Oliver et al., 1997*), *Grpr* KO mice (*Hampton et al., 1998*), *Nmbr* KO mice (*Ohki-Hamazaki et al., 1999*), and their wild-type (WT) littermates were used. We cross *Sst^{Cre}* mice (*Taniguchi et al., 2011*) with a flox-stop channel rhodopsin-eYFP (ChR2-eYFP) line (Ai32) (*Madisen et al., 2012*) to generate mice with ChR2-eYFP expression in *Sst* neurons (Sst-ChR2). All experiments were performed in accordance with the guidelines of the National Institutes of Health and the International Association for the Study of Pain and were approved by the Animal Studies Committee at Washington University School of Medicine.

## Drugs and reagents

The dose of drugs and injection routes are indicated in figure legends. ANP, BNP, CNP, SST, and OCT were supplied from GenScript USA Inc (Piscataway, NJ). GRP$_{18-27}$, NMB and ANP-4–23 were from Bachem (King of Prussia, PA). Histamine, chloroquine, *Npr1* siRNA, *Npr2* siRNA, *Npr3* siRNA, and scrambled control siRNA were purchased from Sigma (St. Louis, MO). BNP-saporin (BNP-sap) and blank-saporin were made by Advanced Targeting Systems. Pertussis toxin (PTX) and gallein were from R&D Systems (Minneapolis, MN) or Selleck (Houston, TX). AP 811 was from Tocris (Minneapolis, MN). U73122 was from Selleck (Houston, TX).

## Behavioral tests

Behavioral tests were videotaped (HDR-CX190 camera, Sony) from a side angle. The videos were played back on the computer and the quantification of mice behaviors was done by persons blinded to the treatments and genotypes.

## Acute scratching behavior

Itch behaviors were performed as previously described (*Sun and Chen, 2007*; *Zhao et al., 2014b*). Briefly, mice were given 30 min to acclimate to the plastic arenas (10 × 10.5 × 15 cm). Mice were then briefly removed from the chamber for drug injections. Injection volume was 10 μL for i.t. injection and 50 μL for i.d. injection. Doses of drugs are indicated in figure legends. The number of scratching responses was counted for 30 min at 5 min intervals. One scratching bout is defined as a lifting of the hind limb toward the body and then a replacing of the limb back to the floor or the mouth, regardless of how many scratching strokes take place between those two movements. Scratching toward the injection site was counted after i.d. injection, and all scratching bouts were counted after i.t. injection.

## Chronic itch models

Dry Skin (Xerosis): The dry skin model was set up as described (*Akiyama et al., 2010*; *Miyamoto et al., 2002*). Briefly, the nape of mice was shaved, and a mixture of acetone and diethyl ether (1:1) was painted on the neck skin for 15 s, followed by 30 s of distilled water application (AEW). This regimen was administrated twice daily for 9 days. Spontaneous scratching behaviors were recorded for 60 min on the morning before AEW treatment. BRAF$^{Nav1.8}$ mice were generated as described previously (*Zhao et al., 2013*).

## RNAscope ISH

RNAscope ISH was performed as described (*Munanairi et al., 2018*; *Wang et al., 2012*). Briefly, mice were anesthetized with a ketamine/xylazine cocktail (ketamine, 100 mg/kg and xylazine, 15 mg/kg) and perfused intracardially with 0.01 M PBS, pH 7.4, and 4% paraformaldehyde (PFA). The spinal cord was dissected, post-fixed in 4% PFA for 16 hr, and cryoprotected in 20% sucrose overnight at 4 °C. Tissues were subsequently cut into 18-μm-thick sections, adhered to Superfrost Plus slides (Fisher Scientific), and frozen at −20 °C. Samples were processed according to the manufacturer's instructions in the RNAscope Fluorescent Multiplex Assay v2 manual for fixed frozen tissue (Advanced Cell Diagnostics), and coverslipped with Fluoromount-G antifade reagent (Southern Biotech) with DAPI (Molecular Probes). The following probes, purchased from Advanced Cell Diagnostics were used: *Nppb* (nucleotide target region 4–777; accession number NM_008726.5), *Sst* (nucleotide target region 18–407; accession number NM_009215.1), *Npr1* (nucleotide target region 941–1882; accession number NM_008727.5), *Npr2* (nucleotide target region 1162–2281; accession number NM_173788.3), *Npr3* (nucleotide target region 919–1888; accession number NM_008728.2), *Grp* (nucleotide target region 22–825; accession number NM_175012.2), *Grpr* (nucleotide target region 463–1596; accession number - NM_008177.2), *Nmbr* (nucleotide target region 25–1131; accession number NM_008703.2), *Vgat* (*Slc32a1*, nucleotide target region 894–2037; accession number NM_009508.2), and *Vglut2* (*Slc17a6*,nucleotide target region 1986–2998; accession number NM_080853.3). Sections were subsequently imaged on a Nikon C2+ confocal microscope (Nikon Instruments, Inc) in three channels with a 20 X objective lens. The criterion for including cells as positive for a gene expression detected by RNAscope ISH: A cell was included as positive if two punctate dots were present in the nucleus and/or cytoplasm. For co-localization studies, dots associated with single DAPI stained nuclei were assessed

as being co-localized. Cell counting was done by a person who was blinded to the experimental design.

## ISH and immunohistochemistry

ISH was performed using digoxigenin-labeled cRNA probes as previously described (*Chen et al., 2001*). Briefly, mice were anesthetized with an overdose of a ketamine/xylazine cocktail and fixed by intracardiac perfusion of cold 0.01 M PBS, pH 7.4, and 4% paraformaldehyde. The spinal cord, DRG, and hairy nape skin tissues were immediately removed, post-fixed in the same fixative overnight at 4 °C, and cryoprotected in 20% sucrose solution. DRGs, lumbar spinal regions and hairy nape skin were frozen in OCT and sectioned at 20 µm thickness on a cryostat. Immunohistochemical (IHC) staining was performed as described (*Zhao et al., 2007*). Spinal cord and DRG tissues were sectioned at 20 µm thickness. Hairy nape skin was sectioned at 30 µm thickness. Free-floating sections were incubated in a blocking solution containing 2% donkey serum and 0.1% Triton X-100 in PBS (PBS-T) for 2 h at room temperature. The sections were incubated with primary antibodies overnight at 4 °C, washed three times in PBS, incubated with the secondary antibodies for 2 hr at room temperature, and washed three times. Sections were mounted on slides with Fluoromount G (Southern Biotech) and coverslips. Fluorescein isothiocyanate (FITC)-conjugated Isolectin B4 from *Griffonia simplicifolia* (IB4, 10 µg/mL; L2895, Sigma), IB4-AlexaFluor 568 conjugate (2 µg/mL, ThermoFisher Scientific) or the following primary antibodies were used: rabbit anti-CGRPα (1:3000; AB1971, Millipore), guinea pig anti-Substance P (1:1000; ab10353, Abcam), guinea pig anti-TRPV1 (1:1000; GP14100, Neuromics), chicken anti-NF-H (1:2000, EnCor Biotechnology, CPCA-NF-H), rabbit anti-βIII-Tubulin (1:2000, Biolegend, 802001), rabbit anti-GFP (1:1000, Molecular Probes, A11122), chicken anti-GFP (1:500, Aves Labs, GFP-1020). The secondary antibodies were purchased from Jackson ImmunoResearch Laboratories, including Cyanine 3 (Cy3), Cyanine 5 (Cy5) - or FITC conjugated donkey anti-rabbit, anti-mouse, anti-chicken or anti-guinea pig IgG (Cy3, 0.5 µg/ml; FITC, 1.25 µg/mL), biotin-SP (long-spacer)-conjugated donkey anti-rabbit IgG (1 µg/mL) and avidin-conjugated Alexa Fluor 488 (0.33 µg/mL). Images were taken using a Nikon Eclipse Ti-U microscope with Cool Snap HQ Fluorescent Camera and DS-U3 Brightfield Camera controlled by Nikon Elements Software (Nikon) or a Leica TCS SPE confocal microscope with Leica LAS AF Software (Leica Microsystems). The staining was quantified by a person blinded to the genotype using ImageJ (version 1.34e, NIH Image) as previously described (*Zhao et al., 2013*). Images were taken using a Nikon C2+ confocal microscope system (Nikon Instruments, Inc) and analysis of images was performed using ImageJ software from NIH Image (version 1.34e). At least 3 mice per group and 10 lumbar sections across each group were included for statistical comparison.

## Small interfering RNA treatment

Negative control siRNA (SIC001) and selective siRNA duplex for mouse *Npr1* (SASI_Mm01_00106966), mouse *Npr2* (SASI_Mm01_00201357), and mouse *Npr3* (SASI_Mm01_00036567) were purchased from Sigma. RNA was dissolved in diethyl pyrocarbonate-treated PBS and prepared immediately prior to administration by mixing the RNA solution with a transfection reagent, RVG-9R (Genscript). The final concentration of RNA was 2 µg/10 µL. siRNA was delivered to the lumbar region of the spinal cord. The injection was given once daily for 6 consecutive days as described previously with some modifications (*Liu et al., 2011*; *Luo et al., 2005*; *Tan et al., 2005*). Behavior testing was carried out 24 hr after the last injection.

## BNP-saporin treatment

Mice were treated with one-time i.t. injection of BNP-sap (2.5 µg/mouse) as previously described (*Mishra and Hoon, 2013*) with the reduced dose, due to the lethal effect of BNP-sap (5 µg). Behavioral tests were performed 2 weeks after BNP-sap injection. After behavioral tests, the spinal cord/DRGs of mice were processed for real-time RT-PCR and ISH.

## Real-time RT-PCR

Real-time RT-PCR was performed as previously described with Fast-Start Universal SYBR Green Master (Roche Applied Science) (*Liu et al., 2011*; *Liu et al., 2014*). All samples were assayed in duplicate (heating at 95 °C for 10 s and at 60 °C for 30 s). Data were analyzed using the Comparative

CT Method (StepOne Software version 2.2.2.), and the expression of target mRNA was normalized to the expression of *Actb* and *Gapdh*. The following primers were used: *Nppb* (NM_008726.4): 5'-gtcagtcgtttgggctgtaac-3', 5'- agacccaggcagagtcagaa-3'; amplicon size, 89 bp; *Sst* (NM_009215.1): 5'- CCCAGACTCCGTCAGTTTCT –3', 5'- CAGCAGCTCTGCCAAGAAGT –3', amplicon size, 87 bp; *Npr1* (NM_008727.5): 5'- tggagacacagtcaacacagc-3', 5'- cgaagacaagtggatcctgag-3'; amplicon size, 70 bp; *Npr2* (NM_173788.3): 5'- tgagcaagccacccactt-3', 5'- agggggccgcagatatac-3', amplicon size, 60 bp; *Npr3* (NM_008728.2): 5'- TGCACACGTCTGCCTACAAT-3', 5'- GCACCGCCAACATGAT-TCTC –3', amplicon size, 138 bp; *Grpr* (NM_008177.2): 5'-TGATTCAGAGTGCCTACAATCTTC-3', 5'-TTCCGGGATTCGATCTG-3'; amplicon size, 71 bp; *Nmbr* (NM_008703.2): 5'- gggggtttctgt-gttcactc –3', 5'- catggggttcacgatagctc –3', amplicon size, 67 bp; *Actb* (NM_007393.3): 5'-TGTTAC-CAACTGGGACGACA-3', 5'-GGGGTGTTGAAGGTCTCAAA-3'; amplicon size, 166 bp; and *Gapdh* (NM_008084.2): 5'-CCCAGCAAGGACACTGAGCAA-3', 5'-TTATGGGGGTCTGGGATGGAAA-3'; amplicon size, 93 bp.

## Cell or neuronal culture, internalization assay, and calcium imaging

Primary culture of spinal dorsal horn neurons was prepared from 5- to 7-day-old C57Bl/6 mice (*Zhao et al., 2014b*) and NMBR WT and KO mice (*Zhao et al., 2014b*). The protocol is essentially the same as previously described (*Munanairi et al., 2018*). After decapitation, laminectomy was performed, and the dorsal horn of the spinal cord was dissected out with a razor blade and incubated in Neurobasal-A Medium (Gibco) containing 30 μL papain (Worthington) at 37 °C for 20 min. Enzymatic digestion was stopped by replacing it with 1 ml Neurobasal-A medium. After washing with the same medium three times, gentle trituration was performed using a flame polished glass pipette until the solution became cloudy. The homogenate was centrifuged at 1500 rpm for 5 min, and the supernatant was discarded. Cell pellet was re-suspended in culture medium composed of Neurobasal-A medium (Gibco, 92% vol/vol), fetal bovine serum (Invitrogen, 2% vol/vol), HI Horse Serum (Invitrogen, 2% vol/vol), GlutaMax (2 mM, Invitrogen, 1% vol/vol), B27 (Invitrogen, 2% vol/vol), Penicillin (100 μg/mL) and Streptomycin (100 μg/mL) and plated onto 12 mm coverslips coated with poly-D-lysine. After three days of culture, neurons were used for calcium imaging as described previously (*Zhao et al., 2014b*). HEK 293 cells were purchased from ATCC (Cat. CRL-1573) with a certificate of analysis confirming cell line identity by STR profiling and confirming lack of mycoplasma contamination. All experiments were carried out on cells cultured for less than ten passages from the purchased stock. HEK 293 cells were grown in Dulbecco's modified Eagle's medium supplemented with 10% fetal bovine serum in a humidified atmosphere containing 5% $CO_2$. Stable HEK293 cell lines were made as described previously (*Liu et al., 2011*). Briefly, cells were transfected with pcDNA3.1/NMBR, pcDNA3.1/NPR1-mCherry, pcDNA3.1/NPR2-mCherry, or pcDNA3.1/NPR3-mCherry by electroporation (GenePulser Xcell, Bio-Rad). Stable transfectants were selected in the presence of 500 μg/ml G418 (Invitrogen). For internalization assay, HEK 293 cells expressing mCherry-tagged receptors were plated on glass bottom dishes coated with poly-D-lysine overnight and imaged every 10 min for 30 min using a Nikon C2+ confocal microscope system (Nikon Instruments, Inc) in the presence of 10 μM BNP. For calcium imaging assay, HEK 293 cells expressing NMBR were plated onto 12 mm coverslips coated with poly-D-lysine. Overnight cell cultures were loaded with Fura 2-acetomethoxy ester (Molecular Probes) for 30 min at 37 °C. After washing, neurons or HEK 293-NMBR cells were imaged at 340 and 380 nm excitation to detect intracellular free calcium (*Zhao et al., 2014b*). PTX (200 ng/ml), gallein (0.1 mM or 2 mM), or AP 811 (0.1 μM) were pre-incubated with NMBR cells for 10 min to block Gαi, Gβγ, or NPRC, respectively.

## Optogenetic activation of cutaneous fibers

Sst-ChR2 mice and wild-type littermates (*Sst-cre*) were used for optical skin stimulation experiments. The nape skin was shaved 3 days prior to stimulation in all mice tested. One day prior to the experiments, each mouse was placed in a plastic arena (10 × 11 X 15 cm) for 30 min to acclimate. For blue light skin stimulation, a fiber optic cable was attached to a fiber-coupled 473 nm blue laser (BL473T8-150FC, Shanghai Laser and Optics Co.) with an ADR-800A adjustable power supply. Laser power output from the fiber optic cable was measured using a photometer (Thor Labs) and set to 15 mW from the fiber tip. An Arduino UNO Rev three circuit board (Arduino) was programmed and attached to the laser via a BNC input to control the frequency and timing of the stimulation (1, 5, 10, or 20 Hz with 10ms on-pulse and 3 s On – 3 s off cycle for 5 min). During stimulation, the mouse was traced

manually by a fiber optic cable with a ferrule tip that was placed 1–2 cm above the nape skin. Videos were played back on a computer for scratching behavior assessments by observers blinded to the animal groups and genotypes.

## Statistical analyses

Values are reported as the mean ± standard error of the mean (SEM). Statistical analyses were performed using Prism 6 (v6.0e, GraphPad, San Diego, CA). For comparison between two or more groups, unpaired, paired two-tailed *t*-test, one-way ANOVA followed by Tukey post hoc tests, or two-way repeated-measures ANOVA followed by Sidak's post hoc analysis, was used. Normality and equal variance tests were performed for all statistical analyses. p < 0.05 was considered statistically significant.

## Acknowledgements

We thank N Maeda for *Npr1*$^{+/-}$ ±, TRC, CDI, and the TGI for RNAi plasmids, and the Chen lab members for comments. The project has been supported by the National Natural Science Foundation of China (82171764, XTL), Guangzhou science and technology project (202102010104, XTL) and the NIH grants R01NS094344, R01 DA037261-01A1 and R01NS113938-01A1 (ZFC).

## Additional information

### Funding

| Funder | Grant reference number | Author |
|---|---|---|
| National Institutes of Health | R01NS094344 | Qing-Tao Meng<br>Xian-Yu Liu<br>Juan Liu<br>Admire Munanairi<br>Devin M Barry<br>Benlong Liu<br>Hua Jin<br>Yu Sun<br>Qianyi Yang<br>Fang Gao<br>Li Wan<br>Jiahang Peng<br>Jin-Hua Jin<br>Kai-Feng Shen<br>Ray Kim<br>Jun Yin<br>Zhou-Feng Chen |
| National Natural Science Foundation of China | 82171764 | Xue-Ting Liu |
| National Institutes of Health | R01 DA037261-01A1 | Qing-Tao Meng<br>Xian-Yu Liu<br>Juan Liu<br>Admire Munanairi<br>Devin M Barry<br>Benlong Liu<br>Hua Jin<br>Yu Sun<br>Qianyi Yang<br>Fang Gao<br>Li Wan<br>Jiahang Peng<br>Jin-Hua Jin<br>Kai-Feng Shen<br>Ray Kim<br>Jun Yin<br>Zhou-Feng Chen |

| Funder | Grant reference number | Author |
|---|---|---|
| National Institutes of Health | R01NS113938-01A1 | Qing-Tao Meng<br>Xian-Yu Liu<br>Juan Liu<br>Admire Munanairi<br>Devin M Barry<br>Benlong Liu<br>Hua Jin<br>Yu Sun<br>Qianyi Yang<br>Fang Gao<br>Li Wan<br>Jiahang Peng<br>Jin-Hua Jin<br>Kai-Feng Shen<br>Ray Kim<br>Jun Yin<br>Zhou-Feng Chen |

The funders had no role in study design, data collection and interpretation, or the decision to submit the work for publication.

## Author contributions

Qing-Tao Meng, Data curation, Formal analysis, Investigation, Writing – original draft; Xian-Yu Liu, Xue-Ting Liu, Juan Liu, Formal analysis, Investigation, Writing – original draft, Writing – review and editing; Admire Munanairi, Hua Jin, Yu Sun, Qianyi Yang, Fang Gao, Li Wan, Jiahang Peng, Jin-Hua Jin, Kai-Feng Shen, Ray Kim, Jun Yin, Investigation; Devin M Barry, Benlong Liu, Investigation, Writing – review and editing; Ailin Tao, Investigation, Supervision; Zhou-Feng Chen, Conceptualization, Investigation, Supervision, Writing – original draft, Writing – review and editing

## Author ORCIDs

Xue-Ting Liu http://orcid.org/0000-0001-5419-198X
Juan Liu http://orcid.org/0000-0003-3343-5521
Zhou-Feng Chen http://orcid.org/0000-0002-6859-7910

## Ethics

All experiments were performed in accordance with the guidelines of the National Institutes of Health and the International Association for the Study of Pain and were approved by the Animal Studies Committee at Washington University School of Medicine. All of the animals were handled according to approved institutional animal care and use committee (IACUC) protocols of Washington University School of Medicine (#20190163).

## Decision letter and Author response

Decision letter https://doi.org/10.7554/eLife.71689.sa1
Author response https://doi.org/10.7554/eLife.71689.sa2

---

# Additional files

## Supplementary files

• Transparent reporting form

## Data availability

All data generated or analyzed during this study are included in the manuscript and supporting file; Source Data files have been provided for Figure 1B, C, Figure 1-figure supplement 1A, B, Figure 1-figure supplement 2G, H, Figure 2A-F, Figure 3A-I, Figure 4F, G, Figure 5F, H, I, Figure 6A-C, and Figure 6-figure supplement 1A, B, E.

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
