## [Editor Report]

The study by Meng et al., reveals how two distinct neuropeptide signals intersect to drive histaminergic itch. They find that the neuropeptides B-type natriuretic peptide (BNP) and neuromedin B (NMB) crosstalk in the spinal cord, whereby BNP facilitates itch behaviors driven by NMB via coupling between the receptors NPRC and NMBR. These results demonstrate a mechanism underlying spinal cord itch processing.

---

## [Decision Letter]

**Decision letter after peer review:**

Thank you for submitting your article "BNP facilitates NMB-mediated histaminergic itch via NPRC-NMBR crosstalk" for consideration by *eLife*. Your article has been reviewed by 3 peer reviewers, one of whom is a member of our Board of Reviewing Editors, and the evaluation has been overseen by Richard Aldrich as the Senior Editor. The reviewers have opted to remain anonymous.

Essential revisions:

Below is a summary of essential revisions, based on comments supplied by all reviewers (see further below). We encourage you to reply to the individual comments to reviewers.

1) Vigorous characterization of the spinal dorsal horn neuron responses to BNP and NMB.

– How many neurons were analyzed for ca^2+^ imaging experiment and how many of them were activated by the agonists (Reviewers #1-2). Sampling size is a major concern and increased numbers are needed to make solid conclusions (Reviewer #2-3).

– Because of the heterogeneity of the spinal cord neurons, it is important to determine what type of neurons are activated by the peptides (Reviewer #2).

2) Validation of quantification of overlap of Npr3 and Nmbr expression with high quality approaches and explanation of how this relates to published datasets (Reviewer #3).

3) Determine functional role of NPRA and NPRC in BNP facilitation of itch (Reviewer #3).

4) Additional approaches to clarify the role of NPRC in itch, including the use of specific receptor agonists and antagonists (Reviewer #1).

5) Clarification of how a Gi pathway can become excitatory in neurons (Reviewer #2).

6) The intracellular signaling pathway proposed related to G β-γ is only supported by a single pharmacological experiment, and more evidence needed (Reviewer #3).

7) Validation of Sst-ChR2 mice for BNP expression in these neurons and that optogenetics can induce neuropeptide release. Furthermore, a clarification of why optogenetic stimulation of Sst-ChR2 mice does not show itch, which does not fit with the main conclusion of the study (Reviewers #1-3).

8) Recommendations to the authors that include supplying additional images/datasets for figures, clarification of why certain results occurred and explanations of interpretations of datasets (See Recommendations to authors, Reviewers #1-3).

*Reviewer #1:*

Recent work has begun to reveal the role of neuropeptides including B-type natriuretic peptide (BNP) and neuromedin B (NMB) in mediating itch. BNP can act through 3 receptors. Here the authors better define the role of these receptors in itch signaling. They find distinct expression for Npr1, Npr2, and Npr3 in the dorsal horn, with different overlaps with Grpr and Nmbr. Npr3 siRNA decreased histamine but not CQ itch. They focused on Npr3 overlap with Nmbr. BNP pre-injection potentiates itch caused by histamine, CQ, and NMB but not GRP. Bnp-Saporin injection ablates Npr1, Npr3 neurons, and histaminergic itch. Calcium imaging of dorsal horn neurons show that BNP could potentiate NMB induced neuron activation, and transfection of HEK cells showed that NPRC could facilitate NMB responses. They then switch to SST-cre/ChR2 mice to target BNP neurons, but optogenetics did not produce itch. These data indicate that BNP may facilitate histaminergic itch by binding NPRC and potentiate signaling of NMBR.

The strength of this study combines several approaches to helps clarify signaling in itch and reveals a potential crosstalk between NPRC and the NMB signaling pathway. However, additional vigor, approaches, and clarifications about results are needed. We also wonder if the authors could further use specific agonists/antagonists of NPRC to prove their mechanism. Major questions:

1) From the literature, ANP should bind to NPRC with higher affinity than BNP (http://pml.medpress.com.pl/ePUBLI/free/PML287-370.pdf). Given the emphasis of the authors on NPRC, it is unclear why ANP intrathecal injection does not cause or modulate, itch whereas BNP intrathecal injection does cause itch. Can the authors please explain?

2) In Figure 1, the authors inject BNP intrathecally and they conclude that there is delayed kinetics of scratching 20-30 min post-injection, which suggests indirect effects of BNP. Can the authors show that GRP or NMB injection shows faster itch kinetics than BNP?

3) There is a NPRC specific agonist, ANP-4-23. Can the authors use this ligand to prove the role of this receptor in facilitating NMB induced itch?

4) The authors utilized Npr3 siRNA injection to determine the role of the NPRC receptor because the Npr3-/- mice have skeletal abnormalities. There is a specific NPRC antagonist – AP811 (https://www.tocris.com/products/ap-811_5498). Can the authors use this antagonist in some experiments to complement siRNA based datasets?

5) In Figure 4A-B, How many neurons and mice were imaged total in this experiment?

6) In Figure 4D-F, how many HEK cells were imaged and over how many fields/experiments?

7) How faithful are SST-ChR2 reporter mice to endogenous expression of Sst and Nppb?

Other Points:

1) Line 270-271: The authors state "these findings indicate BNP-NPRA signaling is not required for the development of chronic itch." That is an overstatement given that dry skin itch is just one model of itch, not all chronic itch.

2) The schematic in Figure 7 seems to have points in there that are not directly proven by this study. The role of PLCbeta in being downstream of Gi signaling of the BNP receptor is not directly addressed by any experiments. We recommend the authors focus specifically on proven points from their study or qualify clearly in the Figure legend points that are speculative.

*Reviewer #2:*

The strength of the study is to use multiple approaches including mouse genetics, behavior, pharmacology, and imaging to reveal a novel spinal pathway regulating histaminergic itch. The authors showed neuropeptide BNP from DRG sensory neurons and its receptor NPRC in spinal cord neurons can facilitate itch transmission by enhancing the action of another neuropeptide NMB and its receptor NMBR. And their cellular studies suggest that the two receptors NPRC and NMBR present in the same spinal cord neurons interact and boost NMBR signaling. The weakness of the study is the lack of vigorous quantification and analysis of cellular analysis. In addition, there are couple of surprising results need to have further experimental evidence to be clarified.

It is unclear that how many cells and neurons have been analyzed for ca^2+^ imaging experiment and how many of them were activated by the agonists? If only a small subset of neurons was activated, how can they make sure the activated neurons expressing the corresponding receptors?

Spinal cord neurons are highly diverse including excitatory and inhibitory interneurons and projection neurons. It is important to determine what type of neurons are activated by the peptides.

On Line 246: It stated "~37% of Grp+ neurons (44.6 {plus minus} 1.9 in Control vs. 22.6 {plus minus} 2.0 in BNP-sap)". Should it be a ~50% ablation instead of 37% ablation? 50% ablation is pretty significant reduction. It is surprising to see there is no reduction in CQ induced itch after the ablation. Need to show the ablation of GRP image in Figure 5.

It is puzzling why Gi pathway became excitatory in neurons. Normally Gi is coupled to an inhibitory pathway to block neuronal activation. This result needs to be further clarified.

The optogenetic activation of SST+ neurons did not evoke any robust scratching which is surprising. Do these neurons really express BNP and if so whether optogenetic activation lead to release of BNP?

*Reviewer #3:*

The manuscript addresses a critical question regarding chemical itch transmission in the spinal cord, what is the relationship between NMBR/GRPR and NPRA/NPRC in histaminergic versus non-histaminergic itch? The authors adopt a multidisciplinary approach to revealing the facilitatory action of BNP and NMB co-mediated histaminergic itch processing in the spinal cord. The experiments are insightful.

There are a few points I wish to raise:

The authors claimed that about half of Npr3+ neurons expressed Nmbr. But there are few overlapping expressions in Figure 1H. Moreover, previous single-cell RNA-Seq results showed that Npr3 (Glut8) and Nmbr (Glut10 and 11) are expressed in different excitatory neuron types in the dorsal spinal cord (Häring et al., Nat Neurosci, 2018). Since the overlapping expression between Npr3 and Nmbr is the fundamental finding in this work, the authors need to validate their conclusion by high-quality histochemistry experiments and/or other approaches. Is Npr1 overlapped with Npr3?

The authors found that BNP pre-treatment enhanced both histaminergic and non-histaminergic itch responses in mice. They need to demonstrate the contribution of NPRA and/or NPRC in BNP-mediated itch facilitation.

It is interesting to study the intracellular signal pathways underlying the coupling between NPRC and NMBR. However, the authors cannot generalize their statement on the role of G β-γ based on one pharmacological experiment. In fact, gallein pre-treatment did not abolish BNP/NMB-mediated elevation of intracellular ca^2+^ concentration. It just led to a ~50% reduction in intracellular ca^2+^ concentration. The authors need more in vitro and in vivo evidence to address the role of G β-γ on BNP-mediated facilitation of histaminergic itch.

The authors claimed that NPRA is dispensable for dry skin itch and neuropathic itch, and tested the development of persistent scratching responses in a dry skin itch model using Npr1 KO mice. However, no evidence in this work shows that NPRA is dispensable for Sst-mediated neuropathic itch. Does the coupling between NPRC and NMBR contribute to the development of chronic itch and neuropathic itch? The experiments using Sst-ChR2 mice are not correlated with the main conclusion of this study.

1. Lines 117-118, although a population may be homogenous through the spinal cord dorsal horn, that does not mean it has no modality-specific function but rather may need to be defined by subpopulations to understand any modality-specific functions.

2. Low-dose BNP-sap did not completely ablate spinal NPRA+ and NPRC+ neurons. Thus the authors cannot rule out the role of NPRA+ or NPRC+ neurons in non-histaminergic itch. And confocal imaging should be required to examine the internalization of NPRs in Figure 5.

3. In Figure1, laminae are not clearly marked, making it difficult to determine distribution pattern. Image sizes and scales are different even for the same probe (D and F for example). Cell density (DAPI staining) in the dorsal spinal cord is distinct in different images. A lower magnification view of the entire dorsal horn with a high magnification inset would be helpful. How many mice and spinal sections are included for cell counting?

4. Figure1, figure legend: For D-I all the descriptions separately describe "double ISH" and "Venn diagram", but for J-M describe both together. Why not either separately state each or state all at the beginning of the legend.

5. Figure 1—figure supplement 1: Panel A, CGRP/IB4 Npr1 KO subpanel – spinal cord is damaged it is suggested to use a different representative image.

6. Figure 3C, low-dose NMB still triggered scratching responses in some mice. The authors need to validate their results. Without a saline alone control, the authors cannot make any statement.

7. Figure 3F, the results are variable. It is suggested to increase the sample size.

8. Figure 3H, 3J, there is only one scale bar shown on the images, but two sizes (20um and 50um) are included in the figure legend.

9. Figure 3H-3K figure legend: "co-expression" should be "co-express".

10. Figure 4B-4C, please add figure legend of mouse background (control, wildtype, Nmbr KO?). The KCl test should be involved in Figure 4B to confirm healthy neurons.

11. Please describe the method for PTX/gallein pre-treatment.

12. Figure 4F, "Galein" should be "Gallein".

13. Figure 5A-5E: Please include figure legends or titles for red and blue signals.

14. Figure 7—figure supplement 1, SST2R for itch inhibition.

---

## [Author Response]

Essential revisions:Below is a summary of essential revisions, based on comments supplied by all reviewers (see further below). We encourage you to reply to the individual comments to reviewers.1) Vigorous characterization of the spinal dorsal horn neuron responses to BNP and NMB.– How many neurons were analyzed for ca^2+^ imaging experiment and how many of them were activated by the agonists (Reviewers #1-2). Sampling size is a major concern and increased numbers are needed to make solid conclusions (Reviewer #2-3).

The detailed information, including sample size, is included in the results and figure legend. Briefly, a total of1513 dissociated dorsal horn neurons isolated from 10 pups were analyzed and 100 responded to 20 nM NMB (~6.6 %) and 16 to 200 nM BNP (~1.1 %). Out of 33 NMBR neurons identified, 8 (~24%) exhibited potentiation by BNP. This number is largely consistent with the finding that approximately 29% of NMBR neurons express Npr3 (Figure 1).

– Because of the heterogeneity of the spinal cord neurons, it is important to determine what type of neurons are activated by the peptides (reviewer #2).

Although spinal cord neurons are heterogeneous, only NMBR neurons responded to certain doses of NMB. Note that the specificity of NMBR neurons was confirmed using NMBR KO neurons. This enabled us to exclude the overwhelming majority of the dorsal horn neurons from our study. Importantly, the finding that co-application of BNP/NMB could evoke calcium transients in neurons co-expressing NPRC and NMBR warrants that only NPRC/NMBR neurons were selected to be examined.

2) Validation of quantification of overlap of Npr3 and Nmbr expression with high quality approaches and explanation of how this relates to published datasets (Reviewer #3).

About 32.0% of Npr3 neurons express Vgat and half of the excitatory Npr3 neurons express Nmbr. That is, about 1/3 of Npr3 neurons overlap with Nmbr neurons, whereas 28.7% of Nmbr neurons express Npr3. These findings are in line with calcium imaging results that 24% of Nmbr neurons responded to co-application of NMB and BNP. In principle, RNAscope ISH is a preferred method to reveal more accurate spatial information for gene expression. For example, RNAseq datasets indicate that all Npr3 transcripts are in Glut8 cluster. However, RNAscope ISH showed that 1/3 of Npr3 neurons express Vgat. It is worth noting that RNAseq datasets should be used as a reference rather than direct evidence due to its well-known limitations and the gene of interests should be validated using either RT-PCR or ISH approaches.

3) Determine functional role of NPRA and NPRC in BNP facilitation of itch (reviewer #3).

Additional pharmacological, behavioral and ca^2+^ imaging data are included to support the function of NPRC-NMBR crosstalk in BNP-facilitation of itch, which is the focus of the present study (Figure 4).

4) Additional approaches to clarify the role of NPRC in itch, including the use of specific receptor agonists and antagonists (Reviewer #1).

NPRC agonist and antagonist data are included (Figure 3 and 4).

5) Clarification of how a Gi pathway can become excitatory in neurons (Reviewer #2).

The role of Gi has been clarified (see Discussion).

6) The intracellular signaling pathway proposed related to G β-γ is only supported by a single pharmacological experiment, and more evidence needed (Reviewer #3).

We show that gallein, a Gβγ inhibitor, significantly inhibits the facilitatory effect of BNP on ca2+ and scratching responses induced by histamine (Figure 4E-G).

7) Validation of Sst-ChR2 mice for BNP expression in these neurons and that optogenetics can induce neuropeptide release. Furthermore, a clarification of why optogenetic stimulation of Sst-ChR2 mice does not show itch, which does not fit with the main conclusion of the study (Reviewers #1-3).

Validation of *Sst*-ChR2 and co-expression of *Sst* and *Nppb* are included (Figure 6 and figure supplement. 1). Technically, it is difficult to determine whether optogenetic activation of *Sst*-ChR2 neurons could release SST/BNP, since endogenous release of inhibitory neuropeptides may not induce overt scratching behavior. Although i.t. SST and BNP could induce scratching behaviors, we argue that they may reflect pharmacological artifacts (see discussion). Our optogenetic results are consistent with previous studies demonstrating that DRG-restricted knockout of SST does not influence itch transmission^1^(also see Discussion).

8) Recommendations to the authors that include supplying additional images/datasets for figures, clarification of why certain results occurred and explanations of interpretations of datasets (See Recommendations to authors, Reviewers #1-3).

See responses above.

Reviewer #1:Major questions:1) From the literature, ANP should bind to NPRC with higher affinity than BNP (http://pml.medpress.com.pl/ePUBLI/free/PML287-370.pdf). Given the emphasis of the authors on NPRC, it is unclear why ANP intrathecal injection does not cause or modulate, itch whereas BNP intrathecal injection does cause itch. Can the authors please explain?

Thanks for raising this important question. We have investigated the role of ANP in itch behaviors. As expected, i.t. ANP failed to evoke robust scratching behaviors. Surprisingly, ANP fails to facilitate histamine-evoked itch (Figure 1 —figure supplement 1). Several possibilities for this intriguing observation have been discussed.

2) In Figure 1, the authors inject BNP intrathecally and they conclude that there is delayed kinetics of scratching 20-30 min post-injection, which suggests indirect effects of BNP. Can the authors show that GRP or NMB injection shows faster itch kinetics than BNP?

The fast response kinetics of i.t. GRP and NMB tests are included in Figure 1—figure supplement 2H.

3) There is a NPRC specific agonist, ANP-4-23. Can the authors use this ligand to prove the role of this receptor in facilitating NMB induced itch?

Thanks for the suggestion. I.t. ANP-4-23 mimics the BNP effect by facilitating i.t. NMB-induced scratching behavior (Figure 3H).

4) The authors utilized Npr3 siRNA injection to determine the role of the NPRC receptor because the Npr3-/- mice have skeletal abnormalities. There is a specific NPRC antagonist – AP811 (https://www.tocris.com/products/ap-811_5498). Can the authors use this antagonist in some experiments to complement siRNA based datasets?

We show that AP 811 mimics Npr3 siRNA effect in both behavior and ca2+ imaging (Figure 3I and 4E-G)

5) In Figure 4A-B, How many neurons and mice were imaged total in this experiment?

Please see the response to essential revision comment #1.

6) In Figure 4D-F, how many HEK cells were imaged and over how many fields/experiments?

The information is added to the figure legend.

7) How faithful are SST-ChR2 reporter mice to endogenous expression of Sst and Nppb?

The data is included in Figure 6F and figure 6 —figure supplement 1.

Other Points:1) Line 270-271: The authors state "these findings indicate BNP-NPRA signaling is not required for the development of chronic itch." That is an overstatement given that dry skin itch is just one model of itch, not all chronic itch.

This has been revised. Several mouse models include the ACD model comprise histaminerigic itch component, and BNP signaling may also be involved.

2) The schematic in Figure 7 seems to have points in there that are not directly proven by this study. The role of PLCbeta in being downstream of Gi signaling of the BNP receptor is not directly addressed by any experiments. We recommend the authors focus specifically on proven points from their study or qualify clearly in the Figure legend points that are speculative.

The data using PLCβ inhibitor U73122 is included (Figure 3I). Both NMBR and GRPR use PLCβ-dependent ca^2+^ signaling that is the canonical signaling pathway (see review^2,3^), downstream of all G_q_ protein-coupled receptors^3^. We have revised the discussion to stress the canonical Gq-PLCβ-Ca^2+^ pathway in the diagram.

Reviewer #2:The strength of the study is to use multiple approaches including mouse genetics, behavior, pharmacology, and imaging to reveal a novel spinal pathway regulating histaminergic itch. The authors showed neuropeptide BNP from DRG sensory neurons and its receptor NPRC in spinal cord neurons can facilitate itch transmission by enhancing the action of another neuropeptide NMB and its receptor NMBR. And their cellular studies suggest that the two receptors NPRC and NMBR present in the same spinal cord neurons interact and boost NMBR signaling. The weakness of the study is the lack of vigorous quantification and analysis of cellular analysis. In addition, there are couple of surprising results need to have further experimental evidence to be clarified.It is unclear that how many cells and neurons have been analyzed for ca^2+^ imaging experiment and how many of them were activated by the agonists? If only a small subset of neurons was activated, how can they make sure the activated neurons expressing the corresponding receptors?

See response to essential revisions #1 and the reviewer 1 above.

Spinal cord neurons are highly diverse including excitatory and inhibitory interneurons and projection neurons. It is important to determine what type of neurons are activated by the peptides.

See response to essential revisions #1.

On Line 246: It stated "~37% of Grp+ neurons (44.6 {plus minus} 1.9 in Control vs. 22.6 {plus minus} 2.0 in BNP-sap)". Should it be a ~50% ablation instead of 37% ablation? 50% ablation is pretty significant reduction. It is surprising to see there is no reduction in CQ induced itch after the ablation. Need to show the ablation of GRP image in Figure 5.

The ablation rate of Grp+ neurons is about 50%. The image is in Figure 5D. In fact, we have ablated about 80% Grp+ neurons in the spinal cord and mice showed normal acute itch behaviors (see Figure 7 Barry et al., 20204). Therefore, spinal Grp+ neurons are dispensable for CQ-induced itch behaviors. The notion that Grp neurons may function as an intact microcircuit has been rebutted in detail recently5.

It is puzzling why Gi pathway became excitatory in neurons. Normally Gi is coupled to an inhibitory pathway to block neuronal activation. This result needs to be further clarified.

The role of Gi has been clarified (see Discussion).

The optogenetic activation of SST+ neurons did not evoke any robust scratching which is surprising. Do these neurons really express BNP and if so whether optogenetic activation lead to release of BNP?

Please see the discussion. Sst and Nppb are largely colocalized (Figure 6 —figure supplement 1).

Reviewer #3:The manuscript addresses a critical question regarding chemical itch transmission in the spinal cord, what is the relationship between NMBR/GRPR and NPRA/NPRC in histaminergic versus non-histaminergic itch? The authors adopt a multidisciplinary approach to revealing the facilitatory action of BNP and NMB co-mediated histaminergic itch processing in the spinal cord. The experiments are insightful.There are a few points I wish to raise:The authors claimed that about half of Npr3+ neurons expressed Nmbr. But there are few overlapping expressions in Figure 1H. Moreover, previous single-cell RNA-Seq results showed that Npr3 (Glut8) and Nmbr (Glut10 and 11) are expressed in different excitatory neuron types in the dorsal spinal cord (Häring et al., Nat Neurosci, 2018). Since the overlapping expression between Npr3 and Nmbr is the fundamental finding in this work, the authors need to validate their conclusion by high-quality histochemistry experiments and/or other approaches. Is Npr1 overlapped with Npr3?

See response to essential revisions #2 and Figure 1 shows that limited overlapping between Npr1 and Npr3.

The authors found that BNP pre-treatment enhanced both histaminergic and non-histaminergic itch responses in mice. They need to demonstrate the contribution of NPRA and/or NPRC in BNP-mediated itch facilitation.

Please see essential revisions #3 and #4.

It is interesting to study the intracellular signal pathways underlying the coupling between NPRC and NMBR. However, the authors cannot generalize their statement on the role of G β-γ based on one pharmacological experiment. In fact, gallein pre-treatment did not abolish BNP/NMB-mediated elevation of intracellular ca^2+^ concentration. It just led to a ~50% reduction in intracellular ca^2+^ concentration. The authors need more in vitro and in vivo evidence to address the role of G β-γ on BNP-mediated facilitation of histaminergic itch.

See response to the reviewer 2.

The authors claimed that NPRA is dispensable for dry skin itch and neuropathic itch, and tested the development of persistent scratching responses in a dry skin itch model using Npr1 KO mice. However, no evidence in this work shows that NPRA is dispensable for Sst-mediated neuropathic itch. Does the coupling between NPRC and NMBR contribute to the development of chronic itch and neuropathic itch? The experiments using Sst-ChR2 mice are not correlated with the main conclusion of this study.

Data from a mouse model of neuropathic itch, in which Nppb and Sst expression are dramatically down-regulated, are added. However, in other chronic itch models with histaminergic itch component, Nppb is upregulated at certain time points6. Therefore, whether BNP is involved in chronic itch depends on the specific mouse models. Sst-ChR2 results are also clarified, which is consistent with the main conclusion of this study.

1. Lines 117-118, although a population may be homogenous through the spinal cord dorsal horn, that does not mean it has no modality-specific function but rather may need to be defined by subpopulations to understand any modality-specific functions.

A neural circuit with modality-specific function in the dorsal horn is usually defined by a gene expression displaying a lamina-specific expression. Spinal modality-specific neural microcircuits with slow response kinetics are proposed to be defined by excitatory GPCR expression in laminae I-II interneurons5. If a gene is additionally expressed in laminae III or other deep laminae, it is unlikely that the population of neurons defined this gene expression is homogenous functionally because lamina III neurons are involved in sensory modalities distinct from laminae I-II interneurons. Even with lamina-specific expression, it is more likely that a GPCR does not define a modality-specific neural circuit, since the number of sensory modalities is limited, while there may be more than a hundred distinct GPCRs expressed in the dorsal horn.

2. Low-dose BNP-sap did not completely ablate spinal NPRA+ and NPRC+ neurons. Thus the authors cannot rule out the role of NPRA+ or NPRC+ neurons in non-histaminergic itch. And confocal imaging should be required to examine the internalization of NPRs in Figure 5.

Yes, we could not exclude a role for NPRA and NPRC neurons in nonhistaminergic itch. Unfortunately, BNP-sap has its limitation because a higher dose required for ablation of NPRA/NPRC neurons completely would kill animals, likely due to its diffusion to the supraspinal region. However, our studies are focused on NPRC-NMBR crosstalk, not on NPRA or NPRC neurons per se. In fact, NMBR neurons are also involved in nonhistaminergic itch as we showed previously, due to their partial activation by GRP as a weak agonist.

Figure 5 are confocal images at high power resolution. Original images are enlarged for better visualization. We used the same method to examine MOR1D/GRPR internalization^7^.

3. In Figure 1, laminae are not clearly marked, making it difficult to determine distribution pattern. Image sizes and scales are different even for the same probe (D and F for example). Cell density (DAPI staining) in the dorsal spinal cord is distinct in different images. A lower magnification view of the entire dorsal horn with a high magnification inset would be helpful. How many mice and spinal sections are included for cell counting?

Thanks for the suggestion. Figure 1 is updated as suggested and n number is included.

4. Figure1, figure legend: For D-I all the descriptions separately describe "double ISH" and "Venn diagram", but for J-M describe both together. Why not either separately state each or state all at the beginning of the legend.

Thanks for the suggestion. We regroup them to avoid redundancy.

5. Figure 1—figure supplement 1: Panel A, CGRP/IB4 Npr1 KO subpanel – spinal cord is damaged it is suggested to use a different representative image.

Replaced.

6. Figure 3C, low-dose NMB still triggered scratching responses in some mice. The authors need to validate their results. Without a saline alone control, the authors cannot make any statement.

A saline control is included in Figure 3 A-D.

7. Figure 3F, the results are variable. It is suggested to increase the sample size.

we increased the sample size (see updated Figure 3F).

8. Figure 3H, 3J, there is only one scale bar shown on the images, but two sizes (20um and 50um) are included in the figure legend.

The scale bar for top panel is 20 µm, while for the lower panel is 50 µm

9. Figure 3H-3K figure legend: "co-expression" should be "co-express".

Corrected.

10. Figure 4B-4C, please add figure legend of mouse background (control, wildtype, Nmbr KO?). The KCl test should be involved in Figure 4B to confirm healthy neurons.

Thanks for the suggestions. Figure 4 legends have been revised as recommended. If neurons responded to 20 nM NMB, it indicates that they are healthy NMBR neurons. KCl is more potent than NMB, and from our experience, all neurons that respond to 20 nM NMB also responded to KCl. However, control is included for Nmbr KO mice (Figure 4C).

11. Please describe the method for PTX/gallein pre-treatment.

The method is included in ca^2+^ imaging section.

12. Figure 4F, "Galein" should be "Gallein".

Corrected.

13. Figure 5A-5E: Please include figure legends or titles for red and blue signals.

Corrected.

14. Figure 7—figure supplement 1, SST2R for itch inhibition.

We clarified the role of SST in itch and pain. SST-SST2R signaling has no role in itch inhibition.

1. Huang, J. et al. Circuit dissection of the role of somatostatin in itch and pain. Nat Neurosci 21, 707-716, doi:10.1038/s41593-018-0119-z (2018).

2. Jensen, R. T., Battey, J. F., Spindel, E. R. and Benya, R. V. International Union of Pharmacology. LXVIII. Mammalian bombesin receptors: nomenclature, distribution, pharmacology, signaling, and functions in normal and disease states. Pharmacol Rev 60, 1-42 (2008).

3. Gilman, A. G. G proteins: transducers of receptor-generated signals. Annu Rev Biochem 56, 615-649 (1987).

4. Barry, D. M. et al. Exploration of sensory and spinal neurons expressing gastrin-releasing peptide in itch and pain related behaviors. Nat Commun 11, 1397, doi:10.1038/s41467-020-15230-y (2020).

5. Chen, Z. F. A neuropeptide code for itch. Nat Rev Neurosci (2021).

6 . Liu, X. T. et al. Spinal GRPR and NPRA contribute to chronic itch in a murine model of allergic contact dermatitis. J Invest Dermatol 140, 1856-1866, doi:10.1016/j.jid.2020.01.016 (2020).

7. Liu, X. Y. et al. Unidirectional cross-activation of GRPR by MOR1D uncouples itch and analgesia induced by opioids. Cell 147, 447-458, doi:10.1016/j.cell.2011.08.043 (2011).